# A Practitioner's Guide to Continual Multimodal Pretraining

**Vishaal Udandarao**[1,3*]   **Karsten Roth**[1,2,6*]   **Sebastian Dziadzio**[1∘]   **Ameya Prabhu**[1∘]
**Mehdi Cherti**[4]   **Oriol Vinyals**[5]   **Olivier Hénaff**[5]
**Samuel Albanie**[†]   **Zeynep Akata**[2,6,7†]   **Matthias Bethge**[1†]

[1]Tübingen AI Center, University of Tübingen [2]Helmholtz Munich [3]University of Cambridge
[4]LAION, Jülich Supercomputing Center (JSC), Research Center Jülich (FZJ), Helmholtz
[5]Google DeepMind [6]Munich Center for ML [7]Technical University of Munich
[*]equal project lead, order interchangeable [∘]core contributors [†]equal supervision.

## Abstract

Multimodal foundation models serve numerous applications at the intersection of vision and language. Still, despite being pretrained on extensive data, they become outdated over time. To keep models updated, research into continual pretraining mainly explores scenarios with either (1) infrequent, indiscriminate updates on large-scale new data, or (2) frequent, sample-level updates. However, practical model deployment often operates in the gap between these two limit cases, as real-world applications demand adaptation to specific subdomains, tasks or concepts — spread over the entire, varying life cycle of a model. In this work, we *complement current perspectives on continual pretraining through a research test bed and offer comprehensive guidance for effective continual model updates in such scenarios*. We first introduce `FoMo-in-Flux`, a continual multimodal pretraining benchmark with realistic compute constraints and practical deployment requirements, constructed over 63 datasets with diverse visual and semantic coverage. Using `FoMo-in-Flux`, we explore the complex landscape of practical continual pretraining through multiple perspectives: (1) data mixtures and stream orderings that emulate real-world deployment settings, (2) methods ranging from simple fine-tuning and traditional continual learning strategies to parameter-efficient updates and model merging, (3) meta-learning-rate schedules and mechanistic design choices, and (4) model and compute scaling. Together, our insights provide a *practitioner's guide to continual multimodal pretraining* for real-world deployment. Benchmark and code is provided here: github.com/ExplainableML/fomo_in_flux.

## 1   Introduction

Foundation models [14] require vast datasets and computational resources to train [142, 29]. Despite these substantial investments, models often have limited concept coverage [180], and quickly become outdated as new tasks emerge. To stay relevant, they need *continual pretraining*, which falls into two high-level categories: (1) infrequent, large-scale updates with substantial new data and compute [49, 76], and (2) frequent, but minimal updates that target specific information, e.g. via knowledge editing or updating knowledge bases in retrieval-augmented systems [28, 191, 135, 57]. However, many real-world applications operate in the large gap between these cases; calling for specialized knowledge (fine-grained expertise or semantic and visual distribution shifts [87, 216, 179, 164, 117, 139, 156, 56, 155, 225, 47, 137]) that goes beyond simple edits, but does not warrant full retraining. Under the semantic versioning framework [143, 140], such specialized minor updates go beyond simple patches, but do not justify major version updates. In this work, we provide a novel framework to emulate practical deployment scenarios for vision-language foundation models in a controlled environment, and study how continual pretraining can succeed:

38th Conference on Neural Information Processing Systems (NeurIPS 2024) Track on Datasets and Benchmarks.

*Creating* `FoMo-in-Flux` (*Foundation-Models-in-Flux*, Fig. 1), which enables a controlled study of *minor* updates of multimodal models over a long life cycle and builds on 63 image classification and retrieval datasets enhanced with captions for multimodal pretraining. Unlike noisy web-crawl datasets (e.g. TiC-RedCaps, DataComp [49, 45]), `FoMo-in-Flux` comprises curated samples with fine-grained class information for precise control of data streams over visual/semantic domains.

***Realistic Continual Pretraining.*** Unlike traditional continual learning, we eschew the *practically irrelevant restriction of limited storage* [135, 136], and allow unrestricted access to pretraining data. As deployment cost is primarily a function of computation, we only impose a restriction on compute budgets. To avoid skewed compute metrics [38, 118], we introduce *Memory-Adjusted FLOPs (MAFs)*, which account for FLOPS in forward and backward passes and peak accelerator memory.

***Methods and Training Recipes for Continual Pretraining.*** Using `FoMo-in-Flux`, we study current research strategies for multiple *minor* continual pretraining updates — from continual learning (CL) regularization strategies (`EWC` [86], `SI` [209]), simple finetuning, parameter-efficient adaptation (`LoRA` [72], `VeRA` [88]), to model merging [78]. We also show the importance of strategies beyond method choices, such as learning rate scheduling, and propose meta schedules for long-term model updates. Moreover, we study model and compute scaling for continual pretraining, and give an overview of important experimental design choices for continual multimodal pretraining pipelines.

***A Data-centric Perspective on Continual Pretraining.*** Concepts and tasks arise in sequence, driven by the ongoing discovery of model shortcomings or desiderata from feedback loops [46]. Our fine-grained control over the sequence of semantic and visual concepts allows us to simulate realistic data streams to better understand how different concept and task orderings affect accumulation of new and retention of existing knowledge. Finally, we provide insights into data mixtures on the accumulation and retention trade-off as new concepts and subdomains are introduced.

Based on our experiments, we collate key practical insights for *continual multimodal pretraining*:

> An Abridged Practitioner's Guide to Continual Multimodal Pretraining.
>
> **1. Method Choices.** Under practical update and compute restrictions, continual learning and parameter-efficient fine-tuning methods favor knowledge retention (stability) while simple fine-tuning focuses on adaptation (plasticity). In combination with **model merging**, fine-tuning sufficiently addresses this trade-off for strong knowledge retention *and* adaptation.
> **2. Meta Learning Rate Schedules.** Learning rates matter, and can naturally be accounted for in long-horizon continual pretraining via **meta learning rate schedules** across incoming tasks; reducing the loss of pretraining knowledge while encouraging high adaptation. Maintaining the same learning rate schedule between pretraining and updates is less important.
> **3. Model and Compute Scaling.** Simple fine-tuning does not scale well with increased compute resources or more frequent updates, unlike fine-tuning with model merging. On the other hand, **increasing model size** helps it acquire new knowledge while retaining its foundational properties, even within the same compute budget.
> **4. Data-centric Stream Orderings.** The **order** in which data updates are applied significantly impacts the model's ability to learn new information and retain its zero-shot capabilities. This is important to account for during deployment. However, when underlying data distributions are the same, models converge to **comparable final performance** across update sequences.
> **5. Data mixture ratio.** The ratio between pretraining-, update-, and buffer data affects the model's final performance, and "IID-fying" knowledge accumulation is crucial: Replaying previous adaptation tasks is more relevant that pretraining replay.

## 2 Categorizing Continual Pretraining: A Versioning Perspective

Traditional continual learning is categorized into class-, domain-, and task-incremental settings [181]. However, continual pretraining benchmarks do not fit these categories, as they exhibit high-overlaps in captions as opposed to disjoint classes [76, 15, 102], and time-varying gradual class/domain shifts [49, 101, 21, 135, 103, 189]. Similarly, continual learning strategies are typically grouped [35, 134] into replay [25, 20], regularization [121, 86, 24], and parameter-isolation methods [224, 3, 227], with recent additions like prompt-tuning [193, 194, 168, 141], fixed-representation [116, 222, 138], and model-mixture methods [114, 78] (see [223] for overview). However, continual

| Benchmark | # Samples | # Tasks | Ordering | Domains | Update Style | Multi-modal | Zero-Shot Retention | Compute-Bound | Data-Mixtures | Real World Stream Variants |
|---|---|---|---|---|---|---|---|---|---|---|
| CORe50 [106] | 165K | 9 | Class-/Data-Inc | Objects | Major | × | × | × | × | × |
| Split-ImageNet [195] | 1.2M | 10 | Class-Inc | Web Images | Major | × | × | × | × | × |
| PTM-Adaptation [173] | 30K-100K | 5-20 | Class-Inc | Web Images | Minor | × | × | × | × | × |
| CLAD [184] | 23K | ~2000 | Time-Inc | Synthetic | Patch | × | × | × | × | × |
| OAK [189] | 326K | ~2000 | Time-Inc | Egocentric | Patch | × | × | × | × | × |
| Inc-PASCAL [119] | 11K | 2-6 | Class-Inc | Web Images | Major | × | × | × | × | × |
| Inc-ADE20K [22] | 20K | 2-6 | Class-Inc | Scene Parsing | Major | × | × | × | × | × |
| StreamingQA [103] | 100K | 6 | Time-Inc | Text | Major | × | × | × | × | × |
| TemporalWiki [82] | 32M | 4 | Time-Inc | Text | Major | × | ✓ | × | × | × |
| CKL [81] | 30K | 2 | Task-Inc | Text | Minor | × | ✓ | × | × | × |
| CTrL [182] | 300K | 100 | Task-Inc | Objects | Major | × | × | × | × | × |
| CLEAR [101] | 7.8M | 10 | Time-Inc | Web Images | Minor | × | × | × | × | × |
| ImageNet2K [136] | 1.2M | 20-200 | Class-/Data-Inc | Web Images | Major | × | × | ✓ | × | × |
| Offline-CGLM [136] | 500K | 20-200 | Time-Inc | Web Images | Major | × | × | ✓ | × | × |
| In1K-P365-LT [62] | 62K | 5 | Class-/Data-Inc | Web Images | Minor | × | × | ✓ | × | × |
| NEVIS [15] | 8M | 79 | Task-Inc | Mixed | Major | × | × | ✓ | × | × |
| CLOC [21] | 39M | 39M | Time-Inc | Geolocation | Patch | × | × | ✓ | × | × |
| CGLM [135] | 500K | 500K | Time-Inc | Landmarks | Patch | × | × | ✓ | × | × |
| CLiMB [171] | 1.3M | 4 | Task-Inc | Mixed | Minor | ✓ | ✓ | × | × | × |
| MTIL [220] | 250K | 5-20 | Class-Inc | Mixed | Minor | ✓ | × | × | × | × |
| Ctl-M2D2 [204] | 6.6B | 160 | Domain-Inc | Text | Minor | × | ✓ | × | × | × |
| TiC-DataComp [49] | 100M/1B/12B | 6 | Time-Inc | Web Images | Major | ✓ | ✓ | ✓ | × | × |
| FoMo-in-Flux (**Ours**) | 2.5M | 20+ | Data-Centric | Mixed | Minor | ✓ | ✓ | ✓ | ✓ | ✓ |

Table 1: `FoMo-in-Flux` **comparison to existing benchmarks** used in continual learning/pretraining studies: it features large timesteps, data-centric streams, provides image-text pairs, a minor-update style, measures zero-shot retention, and is compute-constrained.

foundation model updates are dominated by replay [136, 49], parameter-efficient finetuning [62] and retrieval-augmented methods [185, 135, 57], as traditional methods fail under compute constraints [63, 183, 135], even underperforming simple baselines [138, 116, 136, 215]. Hence, we provide a new categorization suitable for continual pretraining literature. , inspired by the semantic software versioning framework [143]. We believe that different scopes of updates require distinct strategies, indicating that no one solution fits all scenarios (see [198] for a survey, and table 1 for an overview of related benchmarks under the semantic versioning umbrella). We believe foundation models require distinct update strategies, similar to major, minor, and patch updates in software versioning:

**Major Updates.** Large-scale continual pretraining over extensive compute, data, and time resources that substantially alter overall performance. Methods focusing on significant updates [49, 76, 51] consistently employ continual fine-tuning of the model, which has been found to be the primary strategy through extensive comparisons with other works [49, 192, 136, 27]. Currently explored topics include continual LR scheduling [59, 76, 211, 127, 74] to minimize the stability gap [36].

**Patch Updates.** Frequent but minor, targeted updates in which continual fine-tuning leads to poor zero-shot capability retention with little new knowledge gained. These are best managed by continual knowledge editing [28, 191] or sample-wise updates using a fixed backbone [135, 228, 57, 116, 52].

**Minor Updates.** Adaptations to whole subdomains and general concepts out of scope for knowledge edits and major updates. Some examples: updating specific parts of a model with LoRA [62, 12, 109, 196], model merging [78, 174, 187], instruction tuning [64, 218, 26], or incorporating expert knowledge [87, 216, 179, 164, 117, 139, 156, 56, 225, 47, 137]), adding visual reasoning over fine-grained object categories [9, 186, 79, 130, 125, 169], or new domains like sketches [30, 129], drawings [129, 99], or synthetic [19, 115] and medical imagery [77, 41]. Multimodal minor updates can also jointly involve new or infrequently encountered concepts [19, 115], s.a. aforementioned fine-grained expert knowledge, medical terminology or new compositions [80].

# 3 The `FoMo-in-Flux` Benchmark

We introduce `FoMo-in-Flux` (*Foundation-Models-in-Flux*), a benchmark for controlled continual *multimodal* pretraining. Our benchmark extends beyond monolithic pretraining datasets, such as TiC-RedCaps/TiC-DataComp [49], to specialized subdomains with fine-grained control over data streams and adaptation over long horizons. Table 1 extensively compares `FoMo-in-Flux` to related benchmarks, showcasing key features that distinguish it from existing works. For the exact experimental setup, including base models and exact derivation of *MAF* budgets, refer appendix H.1.

## 3.1 Creation

**Breakdown.** `FoMo-in-Flux` consists of 63 classification and retrieval datasets—either publicly available or introduced as part of this work—covering $2.53M$ samples and $23,045$ concepts spanning

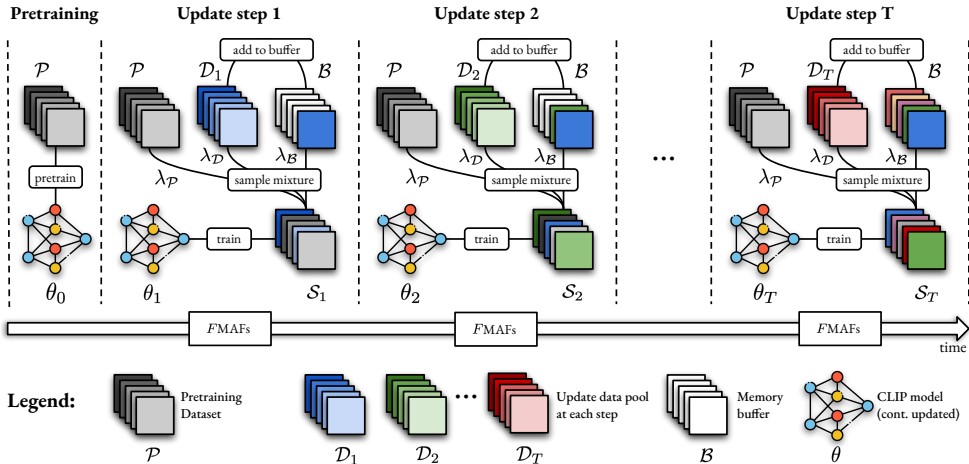

Figure 1: **FoMo-In-Flux pipeline.** *(Pretraining)* We start from pretrained CLIP $\theta_0$ and its pretraining pool $\mathcal{P}$. *(Update steps)* At each step $t$, we sample training instances $\mathcal{S}_t$ from $\mathcal{P}$, current update pool $\mathcal{D}_t$, and memory buffer $\mathcal{B}$ (containing all past $\mathcal{D}_t s$), and train for a fixed compute budget ($F$ MAFs).

diverse visual domains such as natural images, sketches, abstractions, synthetic imagery or generative data. Building concept-first allows experimentation with very precise and controlled ordering on the type of data encountered at each continual pretraining stage. Moreover, by operating on much cleaner data building blocks than web-crawled datasets [49, 45], we ensure cleaner alignment between concepts and images. The 63 datasets are divided into 41 datasets used for *adaptation* only, and 22 hold-out datasets to probe *retention* of initial zero-shot generalization. See tables 2 and 3 for a more detailed overview of datasets, the exact splits, assigned domains and licenses.

***Obscure* Things and Animals.** To improve diversity and to include classes that are systematically seen as long-tailed, we create the *Obscure Animals* and *Obscure Things* datasets using text-to-image models. This also allows us to model the issue of AI-generated content making its way into model training data, potentially misrepresenting some concepts (see *e.g.*, Fig 9). The exact generation is depicted in the supplementary.

**Captioning.** We generate high-quality captions for each image (where needed) through two different methods: **(1)** A scalable two-stage captioning mechanism, which uses BLIP-2 [96] to generate general captions for each image and CapsFusion [207] (T5-XL) to merge and align captions with available information on ground-truth class names (c.f. fig. 7). **(2)** Procedural generation for specific datasets (s.a. Shapes3D [19] and DSprites [115]) using available dataset-specific information to create captions containing e.g. information about object location, orientation, size or shape. These captions are then converted into natural language equivalents at random using GPT-4 [4].

## 3.2 Pipeline, Compute Budgeting and Data Restrictions

**Continual Pretraining Updates.** We illustrate the general `FoMo-in-Flux` training and evaluation pipeline in fig. 1. We start with a model $\theta_0$ trained on a large pretraining dataset $\mathcal{P}$, and an empty buffer $\mathcal{B}$. Within the allocated update budget, at each update step $j \in \{1, 2, \ldots, T\}$, the following happens in order: **(1)** The stream reveals a task update pool of $n_j$ image-text pairs $\mathcal{D}_j = \{(i_k^j, t_k^j)\}_{k=1}^{n_j}$ spanning $\mathcal{C}_j$ concepts. **(2)** We create the training data mixture $\mathcal{S}_j$ by sampling from the pretraining data $\mathcal{P}$, buffer $\mathcal{B}$, and current task data $\mathcal{D}_j$ with respective ratios $\lambda_{\mathcal{P}}, \lambda_{\mathcal{B}}$, and $\lambda_{\mathcal{D}}$, such that $\lambda_{\mathcal{P}} + \lambda_{\mathcal{B}} + \lambda_{\mathcal{D}} = 1$. If samples in $\mathcal{B}$ are insufficient (particularly at the start of task adaptation), we oversample from $\mathcal{D}_j$, with $\lambda_{\mathcal{D}}$ fixed. **(3)** We apply a continual update method $\mathcal{M}$ with a fixed compute budget $F$: $\theta_j = \texttt{train}(\mathcal{M}, \mathcal{D}_j, \theta_{j-1})$. This compute budget $F$ also determines the overall number of update steps conducted. **(4)** We add samples from the update pool $\mathcal{D}_j$ to the unrestricted buffer $\mathcal{B}$. However, while all samples can be stored in buffer $\mathcal{B}$, they cannot all be sampled for training set $\mathcal{S}$, as the compute budget $F$ imposes an implicit memory restriction [136].

**How to Measure Continual Pretraining Computational Cost?** We impose a fixed compute budget for each update step to account for each method's efficiency. Recent works use number of iterations (forward/backward passes) [136, 49], number of updated parameters [88, 13, 118], FLOPs [50], and

throughput [118]. However, a single metric does not paint a complete picture of efficiency [38, 118]. To account for this, we introduce *Memory-Adjusted-FLOPs (MAFs)*, a novel metric that highlights two aspects most relevant from a practitioner's perspective: total number of FLOPs per iteration and maximum utilization of device memory. To compute MAFs, which determines the number of updates a model can take, we multiply FLOPs of each method by the ratio of that method's maximum memory utilization to the maximum memory utilization of a full fine-tuning of the base model.

**Data Restrictions.** We allow unrestricted access to pretraining data (e.g., LAION-400M [160]), and an unlimited replay buffer $\mathcal{B}$, as data storage is a negligible contributor to real-world cost [135, 136], and buffer memory is only utilized during the continual pretraining process. To study different retraining data pools, we use four popular image-text pretraining datasets of varying sizes, quality and curation strategies—LAION-400M [160], CC-12M [23], CC-3M [161], and DataComp-Small [45].

**Metrics & Plots.** We study *adaptation* to new data and *retention* of pretraining knowledge, measured as Knowledge Accumulation ($\mathcal{A}_{KA}$), the avg. accuracy (recall@5 for retrieval) over concepts in all 41 adaptation datasets, and Zero-Shot Retention ($\mathcal{A}_{ZS}$), the zero-shot transfer on the held-out datasets. In most plots, we depict the zero-shot baseline as black star and the joint training upper-bound as golden star. A dotted connection approximates the joint training trajectory on the $\mathcal{A}_{KA}$-$\mathcal{A}_{ZS}$ plane.

### 3.3 Designing Data-Centric Task-Sequences

In addition to studying different pretraining sets $\mathcal{P}$ and data mixture ratios $(\lambda_{\mathcal{P}}, \lambda_{\mathcal{B}}, \lambda_{\mathcal{D}})$, we also investigate different realistic orderings by breaking down FoMo-in-Flux into individual concepts, and then ordering based on a chosen criterion (including an option to reverse orderings). This is visualized in Fig. 10. To do so, having a controlled set of image-caption pairs is critical, as it allows for well-defined and meaningful arrangement of concepts into sequences according to an ordering $\pi(\mathcal{C})$. Each ordering $\pi$ divides the set of samples $\mathcal{D}$ into $T$ disjoint subsets $\{\mathcal{D}_1, \ldots, \mathcal{D}_T\}$ of concepts $\mathcal{C}$ sampled without replacement, i.e. $\mathcal{C}_i \bigcap \mathcal{C}_j = \phi, \forall i, j$. We define six different orderings:
**1. Easy-To-Hard Ordering** (performance) is motivated by curriculum learning [58, 154, 165, 170, 208], assuming users deploying their model to easier concepts and usecases first, with incremental movement towards to harder concepts. **2. Concept Frequency Ordering** (concept-frequency) draws motivation from Udandarao et al. [180], starting from least frequent concepts first (edge cases that are most likely to cause undesired performance drops) and incrementally extending to more frequent concepts represented well in the pretraining pool. **3. Concept Similarity Ordering** (similarity), inspired by Yıldız et al. [204], is based on the hypothesis that training on conceptually similar tasks allows users to minimize catastrophic forgetting over tasks. **4. Time-incremental Ordering** (time), inspired by [15, 73, 21, 135, 49], arranges in chronological order. **5. Dataset-Incremental Ordering** (dataset) is motivated by [148, 111, 112, 190, 206], but extended to a larger sequence of datasets. This sequence is then broken down into the desired number of tasks $T$. **6. Random Ordering** (random) (e.g. [149, 200, 70, 136]) mimics a scenario where user requests for model improvement are unstructured. For this ordering, we simply shuffle class names at random.

## 4 Continual Pretraining: A Method Perspective

We first explore how different methods affect knowledge accumulation and zero-shot retention. We excluded prompt-tuning-based continual learning methods, which often collapse to a single prompt [175] or near-chance performance over a longer time horizon [138]. Similarly, we do not include distillation-based CL methods, as they do not show improvements when memory is unrestricted [136]. For details on each tested method, we refer to the supplementary.

### 4.1 Parameter-efficient Finetuning and Continual Learning

We study *parameter-additive* methods (LoRA [72], VeRA [88], DoRA [104]) and *parameter-selective* approaches tuning only particular weight subsets (LNFit [37], BitFit [8]). We also investigate recently proposed low-rank approximations to model gradient updates (GaLore [219]). We further examine how prior continual learning methods such as Elastic Weight Consolidation (EWC [86]) or Synaptic Intelligence (SI [209]) perform at scale. To begin, we find two extreme points:

**(1) Strongest accumulation, weakest retention.** Naive contrastive finetuning (in orange, fig. 2 left) which achieves strongest knowledge accumulation $\mathcal{A}_{KA}$ across a full update cycle, at the cost

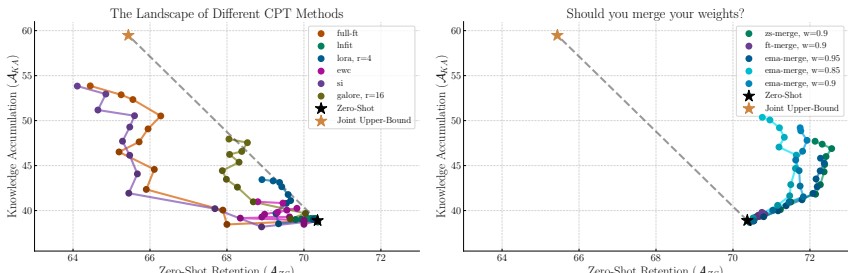

Figure 2: **Which methods to use for continual pretraining over long update cycles?** *(Left)* An in-depth study across five different method families: Continual finetuning (`Full-FT` [76]) and parameter-selective tuning (`LNFit` [37]) provide the extreme points in knowledge accumulation and retention. Switching from `GaLore` [219] to parameter-efficient tuning (`LoRA`) and continual learning methods (`EWC` [86], SI [209]) provides near linear transition points between both extremes. *(Right)* Judiciously merging model weights exhibits unique long-horizon continual pretraining behaviour, allowing for significantly consistent accumulation across update tasks with maximal retention.

of a significant drop in zero-shot retention $\mathcal{A}_{ZS}$ even with learning rate rewarming [76], following best practices sketched out in [54]. We update both the image and language branch, and initialize from the pretraining temperature (c.f. appendix D.3). Importantly, naive finetuning falls victim to "longer-horizon" stability gap issues [36], where forgetting is high and achievable knowledge gain is strongly limited across initial update "steps" (each step being a whole compute-budgeted training cycle, c.f. appendix B.3). **(2) Weakest accumulation, strongest retention.** Parameter-selective methods like `LNFit` (green) and `BitFit` (blue, fig. 11 center) exhibit good knowledge retention, but minimal capacity for the accumulation of new knowledge across longer and complex data streams.

All other methods operate between these end points, trading off knowledge accumulation and retention: **(3) Strong accumulation, weak retention.** By only modifying the naturally lower-rank gradient updates during model training, `GaLore` (olive green, fig. 2 left) offers a moderate balance between the ability to effectively incorporate new knowledge within a given compute budget, and retaining original zero-shot generalization behaviour. **(4) Decent accumulation, decent retention.** Parameter-efficient tuning methods such as `LoRA` (blue, fig. 2 left) and `DoRA` (pink, fig. 11 right) provide an effectively linear reduction in knowledge accumulation and forgetting w.r.t. to base finetuning and `GaLore`. This aligns with recent insights on `LoRA` effectively both learning and forgetting less in single domain finetuning tasks [11]. However, `VeRA` (dark blue, fig. 11 right), which significantly reduces the number of tunable parameters, behaves closely to parameter-selective tuning methods, offering very little knowledge gain across long and complex data streams.

Finally, for continual learning regularization methods we find that while `EWC` (pink, fig. 2 left) significantly improves zero-shot retention, it also offers extremely limited $\mathcal{A}_{KA}$ compared to the initial zero-shot performance. On the other hand, the popular regularisation method `SI` (purple, fig. 2 left) effectively offers no benefits over standard finetuning, either in $\mathcal{A}_{KA}$ or $\mathcal{A}_{ZS}$. The poor performance of regularisation-based methods is curious as prior work has hinted at their benefits at scale [121, 85]. However, our fine-grained, and most importantly compute-controlled `FoMo-In-Flux` helps verify these claims, as these regularization mechanisms are both compute- and memory-expensive.

## 4.2 On the Benefits of Model Merging Techniques

Model merging is a promising avenue for adapting foundation models [197, 78, 172], enabling efficient aggregation of multiple models [203, 157, 34, 5]. Initial work [172] also highlighted potential benefits for small-scale continual learning. To study benefits at scale, we investigate three forms of model merging. Denoting model weights going into task $t$ as $\theta_{t-1}$, finetuned weights after task $t$ as $\theta'_t$, and final model-merged output after task $t$ as $\theta_t$, we define (c.f. fig. 16 for details):
**(1) Exponential-moving averaging** (`EMA-merge`), as adopted in Stojanovski et al. [172], which tunes the previously merged task weights $\theta_{t-1}$ on task $t$ to produce the finetuned weights $\theta'_t$, and then merges $\theta_{t-1}$ with $\theta'_t$ to produce $\theta_t$. **(2) Continual fine-tuning and merging** (`Finetune-merge`) derived from multi-model patching in Ilharco et al. [78]), which produces $\theta_t$ by merging the original pretraining weights $\theta_0$ and the finetuned weights $\theta'_t$. To obtain $\theta'_t$, `Finetune-merge` tunes the previously merged model weights $\theta_{t-1}$, same as `EMA-merge`. **(3) Continual zero-shot merge**

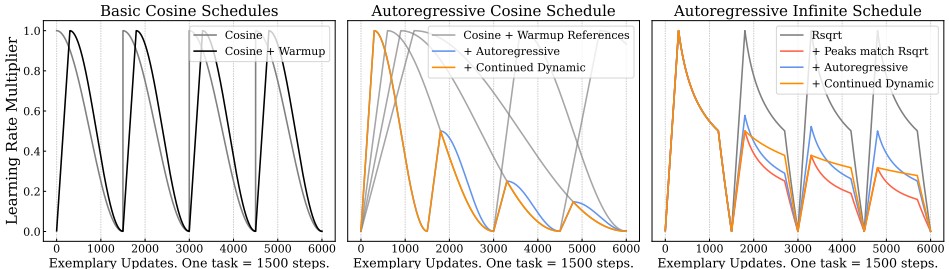

**Figure 3: Visualization of different deployed learning rate schedules**, from task-independent *cosine* and infinite learning rate schedules (*Rsqrt*), to task-dependent meta learning rate schedule.

(`ZeroShot-merge`), a simple ablative merging protocol, which tunes the original pretraining weights $\theta_0$ during each task $t$ and produces $\theta_t$ by merging $\theta_{t-1}$ and the finetuned $\theta'_t$. Each `merge` method uses an old-new weight mixing coefficient $w$, which we ablate over $w=\{0.85, 0.9, 0.95\}$.

As shown in fig. 2 (right), we find that the `EMA-merge` (blue) and `ZS-merge` (green), provide impressive boosts in zero-shot retention rates $\mathcal{A}_{ZS}$ during the first update tasks, and *retain slight gains* over the entire update cycle. Moreover, this is coupled with strong knowledge accumulation $\mathcal{A}_{KA}$, though not yet at the level of standard `finetuning`. As expected, ablating the mixing weight $w$ yields a trade-off between zero-shot retention and knowledge accumulation—higher $w$s provide better zero-shot retention capabilities while compromising on the accumulation $\mathcal{A}_{KA}$. However, across both ablated mixing ratios, as well as the merging mechanism, we find that the high-level continual pretraining dynamics remain the same—at worst limited loss (and at best notable gains) in retention coupled with strong accumulation capacities, while also breaking favorably with the hypothetical linear trade-off between the initial zero-shot performance and the joint finetuning upper-bound!

## 5 Continual Pretraining: General Training Recipes

This section studies the other degrees of freedom orthogonal to methodological update strategies that co-occur within a continual pretraining pipeline: **(1)** The importance of the learning rate and its scheduling in section 5.1 and its translation to meta-learning rate schedules for continual pretraining tasks. **(2)** The impact of both model and compute scaling as independent axes to optimize and account for when planning to deploy a model over longer minor update cycles. More precisely, section 5.2 evaluates the impact on the knowledge accumulation and the zero-shot retention trade-off as a function of both increased model sizes within the same model family, as well as increases in the allocated compute budget within a fixed model size. **(3)** Moreover, the supplementary provides studies on the relevance of locked image and text encoder tuning, as well as the importance of aligning initial and continual pretraining softmax temperature in order to minimize stability gap issues.

### 5.1 Learning Rates, Schedules and Meta-Schedules

By default, LR schedules apply to each task individually [20, 162, 16, 172, 108]. As `open_clip` models use cosine schedules, we first study the impact of re-applying these for each task:

$$\eta_n = \begin{cases} \eta_{\min} + \frac{n}{N_{\text{warm}}}\left(\eta_{\max} - \eta_{\min}\right) & n < N_{\text{warm}} \\ \eta_{\min} + \frac{1}{2}\left(\eta_{\max} - \eta_{\min}\right)\left(1 + \cos\left(\frac{n - N_{\text{warm}}}{N_{\text{task}} - N_{\text{warm}}}\pi\right)\right) \end{cases} \tag{1}$$

with $\eta_n \in [\eta_{\min}, \eta_{\max}]$ the learning rate at step $n$, and $N_{\text{task}}$ the number of update steps for a given task. As recommended in e.g. Ibrahim et al. [76], we utilize linear warmup to the initial pretraining peak learning rate $\eta_{\max}$ used in Cherti et al. [29] for $N_{\text{warm}}$ iterations. To study the impact of a learning rate schedule switch to e.g. infinite learning rate variants for potentially more flexibility down the line, we investigate a switch towards reciprocal square root schedule (*rsqrt*) introduced in Zhai et al. [211]

$$\eta_n = \begin{cases} \eta_{\min} + \frac{n}{N_{\text{warm}}}\left(\eta_{\max} - \eta_{\min}\right) & n \geq N_{\text{warm}} \\ \eta_{\max} \cdot \frac{\sqrt{N_{\text{warm}}}}{\sqrt{n + N_{\text{warm}}}} & n \in [N_{\text{warm}}, N_{\text{task}} - N_{\text{cool}}] \\ \eta_{N_{\text{task}} - N_{\text{cool}}} \cdot \frac{N_{\text{task}} - (n + N_{\text{warm}})}{N_{\text{cool}}} & \text{else} \end{cases} \tag{2}$$

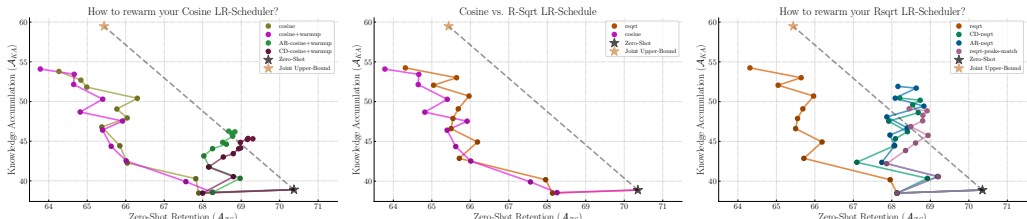

Figure 4: **Meta-scheduling** task-specific LR scheduler has significant impact on the knowledge accumulation and retention trade-off, with meta-schedules derived from infinite LR schedules showing significant transitions across the zeroshot vs finetuning threshold; moving close to accumulation performance of task-independent scheduling, but retaining significantly more pretraining knowledge.

Note that *rsqrt* scheduling includes a separate cooldown section, wherein the last $N_{\text{cool}}$ steps are used to linear cooldown the previously decayed learning rate. Both schedules are visualized in fig. 3 (left and right) over multiple tasks, and the result of either application (matching and changing the pretraining learning rate scheduler) to our 20 task update cycle stream is visualized in fig. 4 (center). As can be seen, there is a negligible change in knowledge accumulation $\mathcal{A}_{KA}$ and knowledge retention for either learning rate scheduler; highlight that across longer update cycles, matching the original pretraining scheduler is of lesser importance.

**Meta Learning Rate Schedules.** By default, each intermediate update is treated independently (c.f. fig. 3 (left)): each task rewarms and cools down the same. However, as these updates appear in succession, catastrophic forgetting of previously seen tasks has to also be accounted for; while with every task update, the model is encouraged to move further away from its pretraining starting point. To reduce the impact of task-level forgetting and the increased shift from pretraining, we introduce meta LR scheduling - task-level schedules over each task-specific, iteration-level LR schedule to account for task continuity. These derive *naturally and hyperparameter-free* by theoretically extending previous task schedule across all the new tasks (see gray hypothetical schedules in fig. 3 (*center*)). We explore four meta-schedules: **(i)** *autoregressive cosine scheduling*, which selects each task $\eta_{\max}$ by building a hypothetical cosine schedule with warmup across all seen tasks and sets it to the intersection point with the warmup process of the current task (Fig. 3 center):

$$\eta_{\max}^T = \eta^{\cos}(n' = N_{\text{warm}}^T + \sum_t^{T-1} N_{\text{task}}^t, N_{\text{task}}' = \sum_t^T N_{\text{task}}^t) \tag{3}$$

where $\eta^{\cos}(\cdot, \cdot)$ defines the LR returned by the standard cosine LR schedule with warmup at point $n'$ for $N_{\text{task}}'$ total iterations. Using the same formulation, we also test **(ii)** *autoregressive continued dynamic* schedule, which warms up to the same $\eta_{\max}^T$, but continues the schedule following the hypothetical cosine schedule over all total previous steps $N_{\text{previous}}$ and the current post-warmup steps $N_{\text{warm}}$. This autoregressive scheduling is naturally extended to the **(iii)** *autoregressive rsqrt schedule*, which sets $\eta_{\max} = \eta^{\text{rsqrt}}(n', N_{\text{task}}')$, and **(iv)** which continues the dynamics of a hypothetically extended base schedule ("*Continued Dynamic*"). Finally, we also introduce **(v)** "*Peaks match Rsqrt*", where respective $\eta_{\max}$ matches the continued dynamics while continuing with a standard rsqrt schedule.

**The impact of task- and meta-level learning rate schedules for continual model updates** are visualized in Fig. 4 on the default 20-task variation of `FoMo-in-Flux` using simple continual finetuning as our reference approach. Indeed, for longer continual pretraining sequences, switching from task-independent to meta learning rate schedules notably changes the accumulation versus retention tradeoff behaviour. While within different meta-schedules variations there is limited difference, as shown in fig. 4 (*left* and *right*), meta-learning rate schedules allow for significantly better retention of initial zero-shot transfer performance. In the case of meta-schedules deriving from cosine learning rate schedules, there is a severe reduction in accumulated new knowledge due to the fast reduction in the learning rate (fig. 3 *left*). Meta-schedules deriving from infinite learning rate schedules like *rsqrt* lend themselves much better to longer-horizon continual pretraining tasks due to the less aggressive decay in learning rate within tasks: As shown in fig. 3 (*right*), the autoregressive *rsqrt* meta-schedule achieves strong gains in $\mathcal{A}_{KA}$, while *vastly increasing the amount of retained knowledge*; exceeding the hypothetical linear zero-shot vs joint finetuning trade-off line.

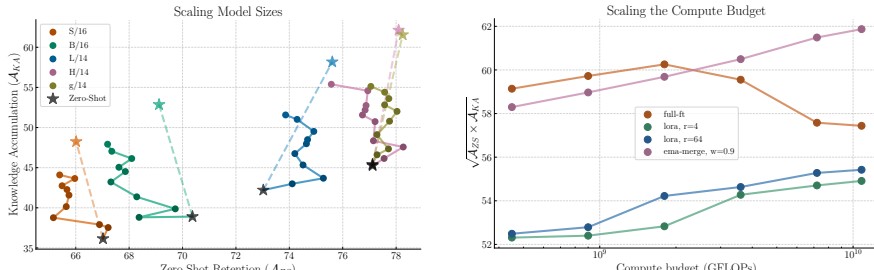

Figure 5: **Model and Compute Scaling for Continual Pretraining.** *(Left)* Increasing model size from ViT S/16 to ViT g/14 scales zero-shot performance consistently. In conjunction however, we find that incorporating new context comes with a *reduced* impact on knowledge retention. *(Right)* For continual finetuning (with/without model merging), as well as LoRA adapters, we consistently increase the allocated compute budget (for B/16). For normal finetuning, an optimum is reached early. With model merging, we instead see a log-linear scaling in performance with additional compute.

## 5.2 Scaling up Model and Compute Budgets

To understand the impact of both model and compute scaling on the ability to continual pretrain over longer update cycles, we adjust either the underlying vision transformer size (keeping the number of update steps and task iterations fxied, and covering ViT-S/16 [$62.3M$], B/16 [$149.6M$], L/14 [$427.62M$], H/14 [$986.11M$] and g/14 [$1366.68M$] taken from [29]) or the allocated compute budget for a fixed model size (selecting our default ViT-B/16 and the default derived finetuning compute budget of $1.8 \times 10^9$ FLOPs as reference). Results for both are provided in fig. 5.

**Scaling Model Size.** As can be seen, we find that with a controlled increase of model size, the ability to continually pretrain over longer minor update cycles improves. While the absolute change in knowledge accumulation $\mathcal{A}_{KA}$ remains rather consistent (within the interval of $8\%$ and $10\%$), zero-shot retention $\mathcal{A}_{ZS}$ improves - where both for the joint finetuning upper bound and continual pretraining, we see improved knowledge retention, and in parts even slight positive backward transfer for ViT-L14 ($3\times$ ViT-B/16). For ViT-B/16, we see $\Delta\mathcal{A}_{KA} \approx 9.0\%$ and negative $\Delta\mathcal{A}_{ZS} \approx 3.2\%$, while for larger L/14, H/14 and g/14 we find ($\Delta_{KA}^{L/14} \approx 9.4, \Delta_{ZS}^{L/14} \approx 0.8$), ($\Delta_{KA}^{H/14} \approx 10.1\%, \Delta_{ZS}^{H/14} \approx -1.5\%$) and ($\Delta_{KA}^{g/14} \approx 9.8\%, \Delta_{ZS}^{g/14} \approx -0.05\%$). Even with higher initial generalization performance, the rate of knowledge accumulation remains roughly the same or even increases, while the ability to maintain its initial generalization capabilities through the longer update cycles in parts *notably improves*. These results suggest that model scaling can benefit long-term re-use and the opportunity to maintain and consistently improve the base model over longer minor update cycles, suggesting model scaling helps mitigate forgetting [145].

**Scaling Compute Budgets.** Instead of investing compute for scaling model size, one can also adjust the directly allocated compute budgets. For our reference model B/16 and its associated compute budget of $1.8 \times 10^9$ FLOPs, we thus study $2\times$, $4\times$ and $6\times$ increases, as well as $0.5\times$ and $0.25\times$ reductions. As seen in fig. 5 (*right*) which aggregates knowledge accumulation $\mathcal{A}_{KA}$ and zero-shot retention $\mathcal{A}_{ZS}$ through their geometric mean, simple continual finetuning (brown) can not consistently leverage increased compute budgets. However, coupled with simple model merging, we find that models become much better at effectively utilizing the additional budget increase; exhibit a log-linear budget-performance relation. With much lower aggregate accumulation-retention performance, we also find a similar, slightly weaker compute scaling behavior for adapter-based continual pretraining.

## 6 Continual Pretraining: A Data-Centric Perspective

This section provides an important data-centric perspective on continual multimodal pretraining (with more information and experiments available in appendix E). We study how fine-grained constraints on the sequence of tasks within an update cycle $\pi$ influence favorable trade-offs between between knowledge accumulation $\mathcal{A}_{KA}$ and zero-shot retention $\mathcal{A}_{ZS}$ (appendix E.3). Results on the impact of different deployment scenarios on continual pretrainability are visualized in fig. 6 for the following scenarios (appendix B.4): **(1)** `performance` sorted - transition from easy to hard concepts, **(2)** `concept-frequency` sorted - rare pretraining concepts first, **(3)** `concept-similarity` sorted -

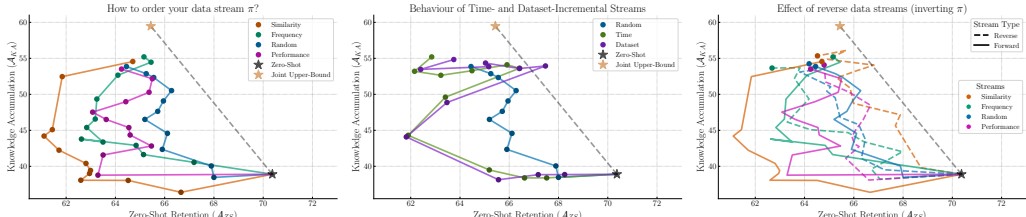

Figure 6: **A Data-centric Perspective on Continual Pretraining.** *(Left)* Four concept-level stream orderings $\pi$ emulating potential update cycles (c.f. section 3.3). Results indicate that deployment scenarios heavily impact intermediate model update stages; however when update cycles operate over shared underlying data distributions, continual pretraining endpoints end up *highly similar*. *(Center)* Dataset-level (random or time-incremental) update cycles exhibit less stable deployment trajectories due to high dataset biases [176, 105].*(Right)* Reversing concept-level datastreams (see appendix E.1) reveals significant trajectory changes. However, the end point similarity still persists.

each update contains concepts semantically related to the preceding update, and **(4)** `random` sorting. `Dataset`-incremental as well as `time`-incremental minor updates are studied separately due to their different structure in section 6, and reverse streams are investigated in section 6.

**Concept- and Sample-based Deployment Scenarios.** Across the deployment scenarios in fig. 6 (left-most), while the `concept-frequency` stream (in green) has the marginally best $\mathcal{A}_{KA}$–$\mathcal{A}_{ZS}$ tradeoff with $\mathcal{A}_{KA}{=}55.2$, $\mathcal{A}_{ZS}{=}65.6$, and `performance` (in pink) performs worst ($\mathcal{A}_{KA}{=}53.8$, $\mathcal{A}_{ZS}{=}64.3$), we find that *convergence end-points are surprisingly similar* - especially w.r.t. the initial zero-shot and the joint finetuning upper bound reference points. However, while endpoints are remarkably similar, different orderings $\pi$ induce significantly different trajectories in the accumulation-retention space, with `similarity` the most sample inefficient ordering, while `random` produces the most favorable trajectories. This aligns with prior work from curriculum learning and active learning that have suggested the efficacy of random curriculums [120, 199], which we find extends itself well into the domain of longer-horizon continual pretraining over minor updates. These insights mean that for longer update trajectories and a shared total space of subdomains and tasks of interest, the type and order of model updates primarily impact initial model versions. This is crucial to account for with respect to the model release horizon and the expected time frame before conducting large-scale continual pretraining updates. However, it also means that across long update horizons irrespective of particular task orders, continually pretrained models arrive at similar performance breakpoints.

**Dataset- and Time-based Deployment Scenarios** differ from the previous scenarios, in that each update step generally contains more semantically grouped samples. As we find for both cases (randomly ordering datasets in `dataset` or time-ordering in `time`), such an update format induces significantly higher trajectory variance, with lesser trajectory coherence when compared to the other four studied streaming orderings. This is expected given prior work suggesting that visual datasets encode heavy biases [176, 105], and hence tasks that explicitly separate these datasets induce larger distribution shifts than tasks that smoothly mix data samples across the datasets on a concept-level. Still, the degree of accumulation remains comparable, though we find zero-shot retention impacted disproportionately higher when orderings $\pi$ or designed on a dataset-level (down to $\mathcal{A}_{ZS} \approx 62.8\%$, compared to $\mathcal{A}_{ZS}^{\texttt{random}} \approx 64.4\%$, $\mathcal{A}_{ZS}^{\texttt{frequency}} \approx 65.5\%$ in the best case). This is important to account for when designing minor updates with the goal of retaining original zero-shot performance.

## 7 Conclusion

This work introduces `FoMo-In-Flux` - a novel, large-scale, fine-grained controllable and long horizon continual pretraining benchmark for vision-language foundation models. Using `FoMo-In-Flux`, we conduct an extensive study into continual multimodal pretraining from a *data-*, *method-*, and *training-centric* perspective. Key findings show that **(1)** model merging strategies successfully trade-off between acquisition of new knowledge and retention of pretraining knowledge, **(2)** learning rates matter; and are well accounted for via meta scheduling, **(3)** that increased model size facilitates inclusion of new knowledge without overwriting pretraining context, **(4)** that simple compute scaling does not benefit all methods equally - with model merging exhibiting the most favorable properties, **(5)** that the order of updates impact the models trajectory in accumulation-retention space, but only marginally impact the streaming endpoints, and that **(6)** replaying on buffer data during streaming is generally more important than replaying on (various subsets of) the original pretraining data.

# Acknowledgements

The authors would like to thank Lukas Thede, Nikhil Parthasarathy, Shashwat Goel and Shyamgopal Karthik for helpful feedback. The authors would also like to thank Kristina Kapanova (University of Tuebingen) for help with infrastructure and compute resources for running all our experiments. VU, KR and SD thank the International Max Planck Research School for Intelligent Systems (IMPRS-IS). VU, KR and SD also thank the European Laboratory for Learning and Intelligent Systems (ELLIS) PhD program for support. VU was supported by a Google PhD Fellowship in Machine Intelligence. SA is supported by a Newton Trust Grant. MB acknowledges financial support via the Open Philanthropy Foundation funded by the Good Ventures Foundation. MB is a member of the Machine Learning Cluster of Excellence, funded by the Deutsche Forschungsgemeinschaft (DFG, German Research Foundation) under Germany's Excellence Strategy – EXC number 2064/1 – Project number 390727645. ZA acknowledges the support from the German Research Foundation (DFG): SFB 1233, Robust Vision: Inference Principles and Neural Mechanisms, project number: 276693517 and ERC Grant DEXIM, project number: 853489.

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

## A  Broader Impact, Limitations and Future Work

**Limitations.**  In this work, our aim was to create a meaningful benchmark, provide practical guidelines, and offer insights into various multimodal continual pretraining scenarios. We focused on *continual, controlled, minor* model updates. We developed `FoMo-in-Flux` to include many publicly accessible datasets covering a wide range of potential adaptation sub-domains. However, our findings on knowledge accumulation $\mathcal{A}_{KA}$ and zero-shot retention $\mathcal{A}_{ZS}$ are tied to our chosen adaptation and evaluation datasets. Consequently, though unlikely, various sub-domains relevant for future applications might not be sufficiently covered. Additionally, our methods were based off of default hyperparameter ranges from original publications (`LoRA`, `VeRA`, `DoRA`, `BitFit`, `LNFit`, `FS-Merge`, `EMA-Merge`) or continual learning repositories (`mammoth` [17]). While we tested the validity of each method and the chosen hyperparameters to elicit meaningful finetuning responses on respective single datasets (as highlighted *e.g.*, for normal full-finetuning in Tab. 5), it overall means that our conclusions rely on the optimality of these provided hyperparameter ranges.

**Broader Impact.**  Better continual model pretraining and the ability to minimize the need for large-scale model retraining can have significant impact on cost, compute and consequently environmental footprint. By encouraging research into extending the re-usability of large-scale pretrained models before a major continual model update or even full retraining from scratch is needed, we believe our work will lead to more economical and ecological utilization of foundation models. We do not believe that there are any immediate negative societal consequences as a result of this work, but we outline the limitations of our datasets in appendix K.

**Future Work.**  Our benchmark and findings provide a crucial starting point reference for further research into continual multimodal pretraining. We sketch a few important and immediate future research directions:

- **(Meta-) Learning Rate Schedules and Beyond:** Our experiments show the importance of learning rate schedules (and meta-variants) designed for longer horizon continual (minor) model updates. We used a default cosine learning rate schedule and one infinite learning rate schedule (rsqrt), along with five meta-schedule variants, but our results showcase that there is a lot of potential in further exploring infinite schedules, as well as extensions into task- and order-conditioned learning rate schedules to allow for continual model pretraining and model updates.
- **Further Scaling Up Compute and Models:** We studied continual learning under realistic constraints (MAFs), with compute budgets derived from `DataComp-small`. Investigating other computational budgets including over-training, and extending budgets to be potentially task-order dependent could have practical relevance. Extending our insights to even larger model scales (ViT-bigG/14 and beyond) can offer further practical guidance. We have investigated the effect of model and compute scaling (see fig. 5) independently and to a first degree, however we believe there is a lot more exciting future work to be done.
- **Text-to-Image Generative Models:** Besides vision-language representation learning, `FoMo-in-Flux` can be used to study continuous minor updates of text-to-image generative models (such as generative diffusion models) on a fine-grained class and concept level, leveraging its diverse set of captions and information about respective image concepts.
- **Optimal Training Mixtures:** Our results indicate that knowledge retention during minor updates depends heavily on replaying data from previous tasks, guided towards "iid"-fying the learning task. This process helps prevent knowledge forgetting related to pretraining. However, there is room to better understand optimal training mixtures within limited compute budgets. Finding the best ways to allocate FLOPs and memory for replay on large pretraining data is crucial.

## B  The `FoMo-in-Flux` Benchmark: Additional details.

### B.1  Creating our *obscure* datasets

We first query ChatGPT to produce a set of 100 obscure animal names and 100 obscure object names. We then ask ChatGPT to produce diverse prompts for each class name to be used as text

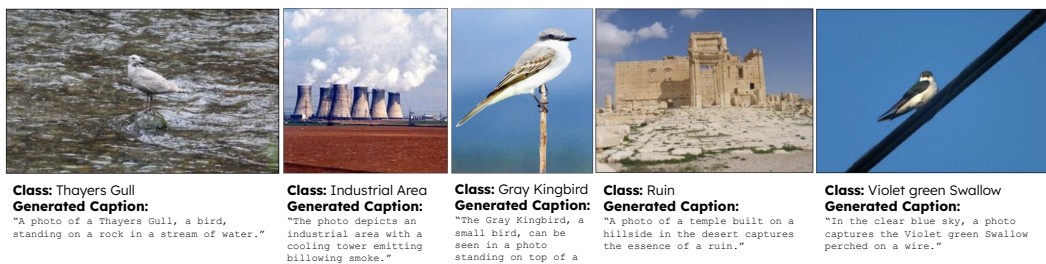

**Class:** Thayers Gull
**Generated Caption:**
"A photo of a Thayers Gull, a bird, standing on a rock in a stream of water."

**Class:** Industrial Area
**Generated Caption:**
"The photo depicts an industrial area with a cooling tower emitting billowing smoke."

**Class:** Gray Kingbird
**Generated Caption:**
"The Gray Kingbird, a small bird, can be seen in a photo standing on top of a tall stem."

**Class:** Ruin
**Generated Caption:**
"A photo of a temple built on a hillside in the desert captures the essence of a ruin."

**Class:** Violet green Swallow
**Generated Caption:**
"In the clear blue sky, a photo captures the Violet green Swallow perched on a wire."

Figure 7: **Visualisation of generated captions.** We showcase some sample captions generated using our two-stage pipeline for fine-grained classes (birds from Birdsnap [9]), and general, coarse classes (taken from SUN397 [202]). The generated captions combine both image descriptions as well as important semantic class information.

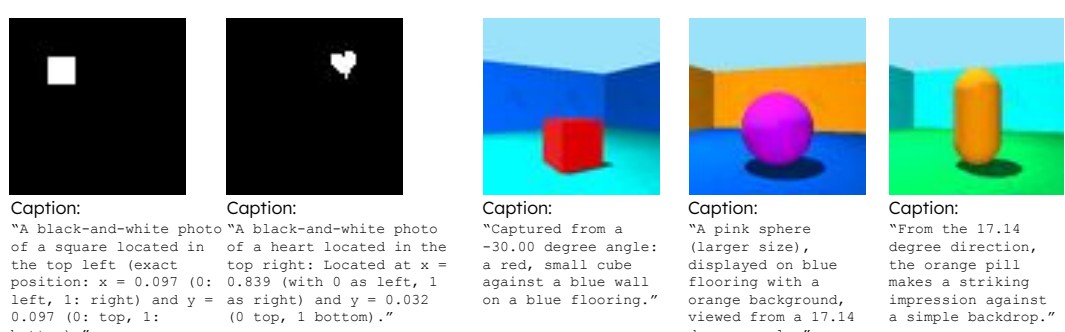

Caption:
"A black-and-white photo of a square located in the top left (exact position: x = 0.097 (0: left, 1: right) and y = 0.097 (0: top, 1: bottom)."

Caption:
"A black-and-white photo of a heart located in the top right: Located at x = 0.839 (with 0 as left, 1 as right) and y = 0.032 (0 top, 1 bottom)."

Caption:
"Captured from a -30.00 degree angle: a red, small cube against a blue wall on a blue flooring."

Caption:
"A pink sphere (larger size), displayed on blue flooring with a orange background, viewed from a 17.14 degree angle."

Caption:
"From the 17.14 degree direction, the orange pill makes a striking impression against a simple backdrop."

Figure 8: **Visualisation of programmatically generated captions** for Shapes3D [19] (*right*) and DSprites [115] (*left*, black and white). Chosen at random, some captions are complete with exact details, while some only have more generic descriptors. Caption style leverages templates generated by GPT-4. The default resolution of these images is $64 \times 64$, hence the low-resolution appearance.

prompts to feed into a text-to-image model. We manually reviewed the quality of text prompts for faithfulness to real world contexts, and then used Kandinsky-2.1 [147], Stable Diffusion-2.1 [153], and Dreamlike-PhotoReal [1] text-to-image models to generate images for each classname using the curated text prompts. Finally, for each class we manually cleaned and filtered the images to ensure faithfulness. We conservatively removed an entire class if more than $30\%$ of its images were ambiguous or unfaithful to the class using reference images from Google Images.

### B.2 Additional information on `FoMo-in-Flux` datasets.

Tables 2 and 3 highlight the diversity of domains and concepts covered in `FoMo-in-Flux`—ranging from diagrams and paintings, natural high- and low-resolution images, to synthetic and generative images, covering fine-grained and specialized domains, such as remote sensingand medical images. On the language side, concept and classes covered also vary noticeably, with e.g. ArtBench10 built around art-style and artist classification (as reflected in the captions), Quilt-1M introducing medical captions for histopathological image data, or our synthetic *Obscure* datasets introducing rare, fantastical concepts with corresponding captions. Dataset licenses are provided in both tables, all of which permit academic re-use. We provide references to original publications, most of which contain information how to download each dataset. To facilitate reproduction, our codebase comes with automatic download mechanisms for datasets where possible, and manual instructions otherwise. Examples for our generated captions are provided in fig. 7 for natural images., and in fig. 8 for procedural generation. Figure 9 contains examples from our generated *obscure* datasets.

### B.3 Pipeline, Compute Budgeting and Data Restrictions - Full Overview.

We illustrate the general `FoMo-in-Flux` training and evaluation pipeline in fig. 1. We start with a model $\theta_0$ trained on a large pretraining dataset $\mathcal{P}$, and an empty buffer $\mathcal{B}$.

Table 2: **Adaptation-only datasets** over various visual and textual domains like diagrams, paintings, natural, synthetic or generative images, remote sensing, art styles, traffic signs or textural data; with datasets from Radford et al. [142] with lower zero-shot performance, common transfer or aggregation benchmark datasets such as DomainNet [129] or VTAB [210] and specialized datasets like MVTec-AD [10].

| Dataset | #Train | #Test | #Classes | Domain | License | Captions |
|---|---|---|---|---|---|---|
| **Classification-based** | | | | | | |
| AI2Diagrams [84] | 2720 | 681 | 15 | diagrams | CC BY-SA | generated |
| ArtBench10 [99] | 47531 | 11883 | 1870 | paintings | Fair Use | generated |
| Birdsnap [9] | 31905 | 7977 | 500 | finegrained, natural | Unspecified, but academic usage | generated |
| Cifar100 [93] | 50000 | 10000 | 100 | natural | Unspecified, but academic usage | generated |
| CLEVR [83] | 55931 | 13983 | 217 | synthetic | CC BY 4.0 | generated |
| CLRS [151] | 13525 | 1475 | 25 | remote sensing | Academic purposes [151] | generated |
| Country211 [142] | 31650 | 21100 | 211 | natural | various CC | generated |
| CUB200-2011 [186] | 5994 | 5794 | 200 | finegrained, natural | custom non-commercial | generated |
| DF20-mini [131] | 32724 | 3637 | 179 | finegrained, natural | custom non-commercial | generated |
| Dollarstreet [152] | 13555 | 4103 | 1701 | finegrained, natural | CC BY-SA 4.0 | generated |
| Domainnet-Clipart [129] | 33525 | 14604 | 345 | illustrations | custom non-commercial | generated |
| Domainnet-Infograph [129] | 36023 | 15582 | 345 | diagrams | custom non-commercial | generated |
| Domainnet-Painting [129] | 50416 | 21850 | 344 | paintings | custom non-commmerical | generated |
| Domainnet-Sketch [129] | 48212 | 20916 | 345 | sketch | custom non-commercial | generated |
| Dsprites [115] | 75000 | 25000 | 27 | synthetic | Apache 2.0 | procedural |
| DTD [31] | 1880 | 1880 | 47 | textural | custom non-commercial | generated |
| FGVCAircraft [110] | 3334 | 3333 | 100 | finegrained, natural | custom non-commercial | generated |
| Flowers102 [125] | 6149 | 1020 | 102 | finegrained, natural | Unspecified, but academic usage | generated |
| FRU92 [69] | 55814 | 9200 | 92 | finegrained, natural | Apache 2.0 | generated |
| iNaturalist2021 [79] | 125000 | 25000 | 2500 | finegrained, natural | custom non-commercial | generated |
| Isicmelanoma [41] | 2245 | 562 | 7 | medical | CC-BY-NC | generated |
| Mitstates [80] | 43002 | 10751 | 1959 | finegrained, natural | Unspecified, but academic usage | generated |
| Mtsd [44] | 59978 | 8737 | 227 | finegrained, traffic signs | CC BY-NC-SA 4.0 | generated |
| MVTec-AD (Base) [10] | 2903 | 726 | 15 | high-resolution, industrial | CC BY-NC-SA 4.0 | generated |
| MVTec-AD (Faults) [10] | 1380 | 345 | 88 | high-resolution, industrial | CC BY-NC-SA 4.0 | generated |
| ObjectNet [7] | 40134 | 10000 | 313 | natural | CC BY 4.0 | generated |
| Obscure Animals | 17000 | 4238 | 74 | generative | MIT | custom |
| Obscure Things | 19128 | 4758 | 84 | generative | MIT | custom |
| OpenImages [90] | 115333 | 8593 | 589 | natural | Apache 2.0 | available |
| PatternNet [226] | 26600 | 3800 | 38 | remote sensing | custom non-commercial | generated |
| Places365 [221] | 120231 | 36499 | 365 | natural | custom non-commercial | generated |
| Plantvillage [75] | 43444 | 10681 | 38 | finegrained, natural | CC0 | generated |
| Quilt-1M [77] | 95862 | 23966 | 157 | medical | Academic purposes | available |
| Resisc45 [68] | 18900 | 6300 | 45 | remote sensing | Unspecified, but academic usage | generated |
| Shapes3D [19] | 75000 | 25000 | 864 | synthetic | Apache 2.0 | procedural |
| SnakeCLEF2023 [130] | 151031 | 14117 | 1599 | finegrained, natural | custom non-commercial | generated |
| SUN397 [202] | 15880 | 19850 | 397 | natural | custom non-commercial | generated |
| SynthCLIP106 [60] | 84800 | 13886 | 106 | generative | CC BY-NC 4.0 | generated |
| Veg200 [69] | 61117 | 20000 | 200 | finegrained, natural | Apache 2.0 | generated |
| Zappos50k [205] | 37829 | 9458 | 1847 | finegrained, object | custom non-commerical | generated |
| **Retrieval-based** | | | | | | |
| FSCOCO [30] (avg T2I/I2T R@5) | 7105 | 1777 | 115 | sketch | CC BY-NC 4.0 | Available |
| **Total** | **1759782** | **453020** | **18449** | | | |

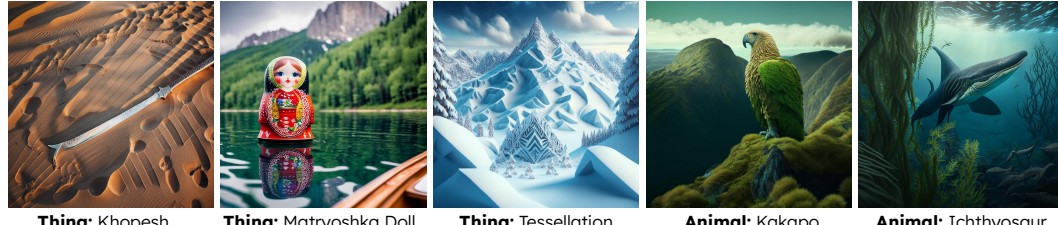

**Thing:** Khopesh    **Thing:** Matryoshka Doll    **Thing:** Tessellation    **Animal:** Kakapo    **Animal:** Ichthyosaur

Figure 9: **Examples of our generated obscure things and animals along with captions**, covering 100 rare and uncommonly occurring things and animals. For each class, images are generated using either Kandinsky-2.1 [147], Stable Diffusion 2.1 [153] or Dreamlike-PhotoReal [1].

**Continual Pretraining Updates.** Within the allocated update budget, at each update step $j \in \{1, 2, \ldots, T\}$, the following happens in order:

1. The stream reveals a task update pool of $n_j$ image-text pairs $\mathcal{D}_j = \{(i_k^j, t_k^j)\}_{k=1}^{n_j}$ spanning $\mathcal{C}_j$ concepts.

2. We create the training data mixture $\mathcal{S}_j$ by sampling from the pretraining data $\mathcal{P}$, buffer $\mathcal{B}$, and current task data $\mathcal{D}_j$ with respective ratios $\lambda_{\mathcal{P}}, \lambda_{\mathcal{B}}$, and $\lambda_{\mathcal{D}}$, such that $\lambda_{\mathcal{P}} + \lambda_{\mathcal{B}} + \lambda_{\mathcal{D}} = 1$.

Table 3: `FoMo-in-Flux` **Evaluation-only Datasets.** We utilize a subset of standard evaluation datasets used in Radford et al. [142], as well as an array of ImageNet-like variations (including the original ImageNet) to probe different aspect of vision-language understanding and alignment. Moreover, datasets like Food101 [18] or OxfordPets [126] were selected due to their high initial zero-shot performance scores.

| Dataset | # Train | # Test | # Classes | Domain | License | Captions |
|---|---|---|---|---|---|---|
| **Classification-based** | | | | | | |
| Caltech101 [94] | 6026 | 2651 | 101 | natural | CC BY 4.0 | generated |
| Caltech256 [55] | 21307 | 9300 | 257 | natural | CC BY 4.0 | generated |
| Cars196 [169] | 8144 | 8041 | 196 | finegrained, natural | custom non-commercial | generated |
| Cifar10 [91] | 50000 | 10000 | 10 | natural, low-res | Unspecified, but academic usage | generated |
| Domainnet-Quickdraw [129] | 60375 | 25875 | 345 | sketch | custom non-commercial | generated |
| EuroSAT [65] | 18900 | 8100 | 10 | Remote Sensing | MIT | generated |
| FashionMNIST [201] | 60000 | 10000 | 10 | b&w, low-res | MIT | generated |
| Food101 [18] | 75750 | 25250 | 101 | finegrained, natural | Unspecified, but academic usage | generated |
| GTSRB [71] | 18635 | 8005 | 43 | traffic signs | CC0 | generated |
| ImageNet [39] | 0 | 50000 | 1000 | natural | custom non-commercial | generated |
| ImageNet-A [67] | 0 | 7500 | 200 | adversarial, natural | MIT | generated |
| ImageNet-D [214] | 0 | 4835 | 103 | generative | MIT | generated |
| ImageNet-R [66] | 0 | 30000 | 200 | renditions (e.g. sketch, paintings) | MIT | generated |
| ImageNet-S [188] | 0 | 50889 | 1000 | sketch | MIT | generated |
| ImageNet-V2 [150] | 0 | 10000 | 1000 | natural | MIT | generated |
| MNIST [40] | 60000 | 10000 | 10 | b&w, low-res | CC BY-SA 3.0 | generated |
| Monkeys10 [2] | 1097 | 272 | 10 | natural | CC0 | generated |
| OxfordPets [126] | 3680 | 3669 | 37 | natural | CC BY-SA 4.0 | generated |
| STL10 [32] | 5000 | 8000 | 10 | natural, low-res | custom non-commercial | generated |
| SVHN [122] | 73257 | 26032 | 10 | natural, low-res | custom non-commercial | generated |
| **Retrieval-based** | | | | | | |
| MSCOCO [100] (avg T2I/I2T R@5) | 0 | 5000 | 0 | natural | CC BY 4.0 | available |
| Flickr30k [132] (avg T2I/I2T R@5) | 0 | 1000 | 0 | natural | CC0 | available |
| **Total** | **462171** | **314419** | **4653** | | | |

If samples in $\mathcal{B}$ are insufficient (particularly at the start of task adaptation), we oversample from $\mathcal{D}_j$, with $\lambda_\mathcal{D}$ fixed.

3. We apply a continual update method $\mathcal{M}$ with a fixed compute budget $F$: $\theta_j = \texttt{train}(\mathcal{M}, \mathcal{D}_j, \theta_{j-1})$. This compute budget $F$ also determines the overall number of update steps conducted.

4. We add samples from the update pool $\mathcal{D}_j$ to the unrestricted buffer $\mathcal{B}$. However, while all samples can be stored in buffer $\mathcal{B}$, they cannot all be sampled for training set $\mathcal{S}$, as the compute budget $F$ imposes an implicit memory restriction [136].

**How to Measure Continual Pretraining Computational Cost?** To keep our setting practical and ensure a fair comparison, we impose a fixed computation cost budget for each time step to account for the efficiency of each method. However, there is no universally adopted measure of computational cost. Recent works use the number of iterations (forward/backward passes) [136, 49], number of parameters updated [88, 13, 118], FLOPs [50], and time/throughput [118]. However, a single metric does not paint a complete picture of efficiency that is relevant in practice [38, 118].

To account for this, we introduce *Memory-Adjusted-FLOPs (MAFs)*, a novel metric that highlights two aspects most relevant from a practitioner's perspective: the total number of FLOPs per iteration and the maximum utilization of device memory. To compute MAFs, we multiply the FLOPs count of each method by a *memory multiplier*, the ratio of that method's maximum memory utilization to the maximum memory utilization of a full fine-tuning of the base model. The total amount of MAFs for each method and backbone determines the allowed number of update steps each method can take during each adaptation task.

**Data Restrictions.** We allow unrestricted access to pretraining data (e.g., `LAION-400M` [160]), and an unlimited replay buffer $\mathcal{B}$, as data storage is a negligible contributor to real-world cost [135, 136], and buffer memory is only utilized during the continual pretraining process. To study different retraining data pools, we use four popular image-text pretraining datasets of varying sizes, quality and curation strategies—`LAION-400M` [160], `CC-12M` [23], `CC-3M` [161], and `DataComp-Small` [45].

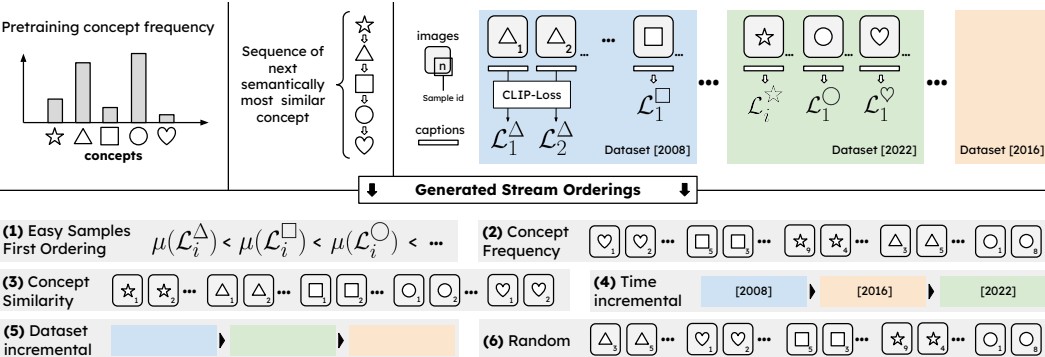

Figure 10: **Pictographic visualization of different data stream orderings** included within the `FoMo-in-Flux` benchmark setup.

## B.4 Designing Data-Centric Task-Sequences

In addition to studying different pretraining sets $\mathcal{P}$ and data mixture ratios $(\lambda_{\mathcal{P}}, \lambda_{\mathcal{B}}, \lambda_{\mathcal{D}})$, we also investigate different realistic orderings by breaking down the `FoMo-in-Flux` datasets into individual concepts, which are then ordered according to a chosen criterion (including the option to study reverse orderings). This is visualized in Fig. 10. In order to do so, having a controlled set of image-caption pairs is critical, as it allows for well-defined and meaningful arrangement of concepts into sequences according to an ordering $\pi(\mathcal{C})$. Each ordering $\pi$ divides the set of samples $\mathcal{D}$ into $T$ disjoint subsets $\{\mathcal{D}_1, \ldots, \mathcal{D}_T\}$ of concepts $\mathcal{C}$ sampled without replacement, i.e. $\mathcal{C}_i \bigcap \mathcal{C}_j = \phi, \forall i, j$. We define and motivate six different orderings below:

**1. Easy-To-Hard Ordering** (`performance`) is motivated by curriculum learning [58, 154, 165, 170, 208], assuming users deploying their model to easier concepts and usecases first, with incremental movement towards to harder concepts.

*Implementation.* We approach the notion of "easy" vs. "hard" samples by ordering them according to base model performance. For each concept, we select 50 random image-text pairs and then randomly sample further 50 image-text pairs from the CC-3M dataset to represent random samples from CLIP's pretraining data pool [29]. For each of the 100 image-text pairs, we compute the sample-wise contrastive loss using a CLIP ViT-L-14 model, and average it over concepts. The lower the mean loss per concept, the easier it is. We then sort all the concepts by their mean loss in ascending order, and consider that to be the data stream ordering.

**2. Concept Frequency Ordering** (`concept-frequency`) draws motivation from Udandarao et al. [180], with user requests for model improvement starting from least frequent concepts first (as these constitute edge cases that are most likely to cause undesired performance drops) and incrementally extending to more frequent concepts, which are already represented well in the pretraining pool.

*Implementation.* We use the *What's In My Big Data* [43] tool's elastic search index to search for the frequency of occurrence of each of the class names in the C4 [144] dataset. We compute the frequencies of each of the classes, and order them such that the least frequent concepts (long-tail) occur first and the most frequent ones (head-concepts) are at the end.

**3. Concept Similarity Ordering** (`similarity`), inspired by Yıldız et al. [204], is based on the hypothesis that training on conceptually similar tasks allows users to minimize catastrophic forgetting over tasks.

*Implementation.* To find a *trajectory* with the highest semantic similarity between subsequent concepts, we start with a similarity matrix containing the pairwise similarities between all the class names (via CLIP ViT-L-14 text embeddings of templated text captions of the respective classes). Defining each class as a node in a graph, with weights between the classes being their similarity, the problem reduces to finding the minimum spanning path. We use a simple greedy algorithm: pick a starting class, find its closest neighbour from the remaining set of classes, and keep repeating until

we exhaust all classes. We repeat this procedure for every class as a starting point and pick the path with the smallest total weight across all starting classes.

**4. Time-incremental Ordering** (`time`), inspired by [15, 73, 21, 135, 49], arranges in chronological order.

*Implementation.* As we only have reliable time information about datasets (via release dates of corresponding publications or the official dataset upload date), concepts are ordered on a dataset-level [15]. These year-level groups are arranged from oldest to most recent, assuming that older datasets are more likely to be conceptually integrated within the pretraining data. Within each year, concepts are randomly ordered. Alongside the above orderings, we compare with two baseline methods popular in continual learning, to better understand the trade-offs made by these data-centric orderings:

**5. Dataset-Incremental Ordering** (`dataset`) is motivated by [148, 111, 112, 190, 206], but extended to a larger sequence of datasets. To set up `dataset`, we simply randomly sample datasets from Tab. 2 to create a dataset-incremental concept sequence. This sequence is then broken down into the desired number of tasks $T$.

**6. Random Ordering** (`random`), a baseline class-incremental ordering widely used across continual learning setups [149, 200, 70, 136], mimics a scenario where user requests for model improvement are unstructured. For this ordering, we simply shuffle class names at random.

### B.5 Verifying Downstream Datasets: Finetuning must improve Performance

In order to estimate a reference upper bound on adaptation performance, verify the quality of generated captions, and perform a sanity-check on our training pipeline, we fine-tune CLIP-ViT-B/32 and CLIP-ViT-B/16 individually on each dataset in our training split, as well as all the evaluation-only datasets which come with training samples. We fine-tune the models on each dataset for 10 epochs, with exact results and training details shown in Supp. table 5. For *all datasets*, we find that finetuning a pretrained CLIP model on our generated captions consistently, and in parts very significantly, improves initial zero-shot performance. This showcases the validity of our generated captions, and supports the inclusion of each listed dataset in the `FoMo-in-Flux` benchmark.

## C Continual Pretraining: Additional Details to our Method Perspective

### C.1 Detailed Method Overview.

In detail, we study several promising directions for continual pretraining of foundation models: *(1) Naive continual finetuning* [49, 136, 76], which has emerged as a dominant approach for major updates on realistic large-scale benchmarks, making it a contender for handling minor updates as well. *(2) Parameter-efficient tuning methods* like `LoRA` [72], which have become a method of choice for minor updates on a smaller scale or for adapting to new tasks with reduced memory requirements [62, 109, 196, 166, 167, 48, 98] through the use of low-rank weight approximations. In a related fashion, recent work by Zhao et al. [219] has shown promise for model finetuning through low-rank approximations on the optimization gradients (`GaLore`). *(3) Parameter-selective tuning methods* such as `BitFit` [8] or `LNFit` [37], which only tune and update particular parameter subsets in the pretrained model such as bias or normalization terms. *(4) Traditional regularization strategies* from continual learning literature [86, 209], which have yielded surprisingly strong performance in recent studies both in parameter [95, 217] and feature space [121], despite being developed and tested in small-scale scenarios where the model is trained from scratch. *(5) Model merging*, which has gained popularity [197, 78, 146] in non-continual learning scenarios as a means to aggregate models tuned across different tasks, and has been studied in some recent [172, 114] and concurrent works [89, 113] as a method to facilitate continual pretraining over longer adaptation periods. All model merging variations are visualized in fig. 16.

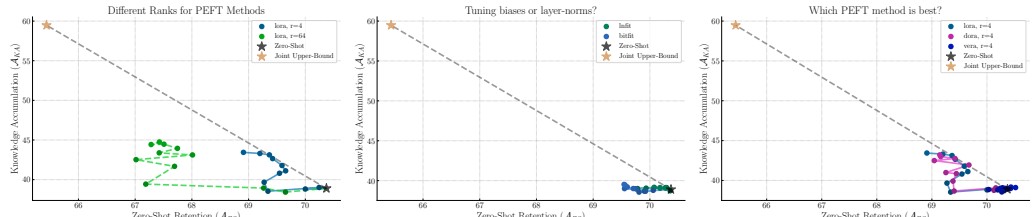

Figure 11: **More Detailed Method Ablations.** (***Left***) Impact of different ranks on continual pretrain-ability; favouring lower rank values ($r = 4$) over large rank values ($r = 64$) when contrasted against the hypothetical linear tradeoff line between original zero-shot behaviour and performance when finetuned over all data at once. (***Center***) Comparison between parameter-selective LNFit [37] and BitFit [8]. Both exhibit similar behaviour: strongly limited ability to continuously incorporate new context, with correspondingly minimal deviation in original zero-shot behaviour. (***Right***) Overview of adaptation versus evaluation trajectories for different PEFT methods: LoRA [72], DoRA [104] and VeRA [88]. LoRA and DoRA behave comparably, with low adaptable parameter counts in VeRA heavily limiting the ability to accumulate new knowledge.

## C.2 Additional study on parameter-efficient finetuning methods.

For parameter-efficient tuning, the scaling between the accumulation-retention trade-off and the tunable parameter count is also unsurprisingly reflected when adjusting the rank of LoRA (fig. 11 left)—though the loss in original generalization performance outweighs the achievable knowledge accumulation when contrasted against the hypothetical trade-off line between initial zero-shot behaviour and joint finetuning.

# D  Continual Pretraining: Additional Details to General Training Recipes

## D.1  On the Influence of Learning Rate Choices for Continual Pretraining.

To define the learning rate of choice for our continual pretraining problem, we derive it directly from the original pretraining values in Cherti et al. [29] (*1e-3*). We note that the exact peak values are corrected for our practical differences in compute availability (operating on a batch-size of $b_{\text{ours}} = 512$ instead of $b_{\text{openclip}} = 88064$; testing both the commonly utilized linear resizing [53]: $\lambda_{\text{scaled}} = b_{\text{ours}}/b_{\text{openclip}} \cdot \lambda_{\text{openclip}}$ and the respective square-root resizing [92] (giving $5.81e-6$ and $7.625e-5$, respectively). In preliminary experiments, we found that rounding up the linearly resized reference (to $\lambda_{\text{scaled}} = 1e-5$) worked slightly better than both options, and provides a much cleaner entry point. As such, we chose to utilize $1e-5$ as our learning rate reference value. As we find in fig. 12, this (mostly) direct

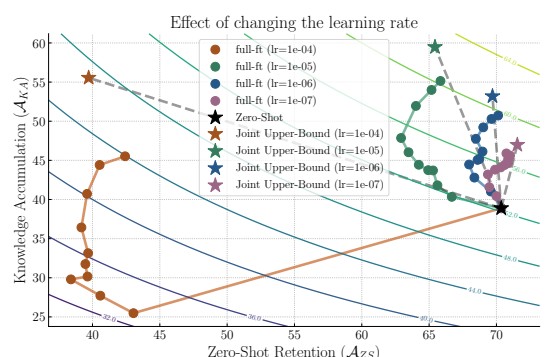

Figure 12: **The effect of the base learning rate on continual pretraining**. The learning trajectory is shown for each value of the learning rate, with the joint training performance as an upper bound. The contour lines show the geometric mean of knowledge accumulation and zero-shot retention ($\sqrt{\mathcal{A}_{KA} \times \mathcal{A}_{ZS}}$). A learning rate of $1e-5$ derived from the inital pretraining learning rate achieves the highest final knowledge accumulation and provides the optimal balance between $\mathcal{A}_{KA}$ and $\mathcal{A}_{ZS}$.

re-use of the maximum learning rate has most importantly the highest degree of knowledge accumulation, but also achieves the highest base joint tradeoff with respect to zero-shot retention. Larger learning rates incur significantly higher rates of particularly early-task forgetting, while smaller learning rates limit the amount of knowledge gained. As such, we set $\lambda_{\text{scaled}} = 1e-5$ as our base learning rate.

## D.2  Model-specific tuning choices in compute-restricted scenarios

Finally, we highlight the relevance of freezing either image or text encoder in practically compute-restricted continual pretraining in Fig. 13. As freezing either the image or language encoder can allow for significant increases (over a magnitude) in the tuning step budget (as total FLOPs and memory use go down), we find that within the compute-restricted continual multimodal pretraining scenario, tuning both encoders still remains beneficial (aligning with insights provided in Goyal et al. [54] for simple finetuning). While there is negligible difference when freezing each encoder respectively (despite the substantial difference in FLOPs reduction based on tuning the image-encoder alone vs. tuning the text-encoder alone), updating the vision-language model as a joint system incurs a more favorable trade-off between knowledge accumulation and zero-shot retention for each update.

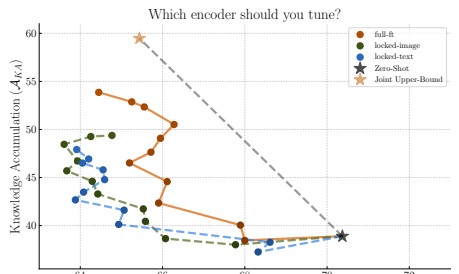

Figure 13: **To freeze or not to freeze.** Tuning both encoders beats single encoder tuning in line with finetuning insights from Goyal et al. [54].

## D.3  Softmax Temperatures for Contrastive Losses—*Not Too Hot*!

Recall that CLIP's contrastive loss uses a temperature parameter $\tau$, and it is typically learnable during pretraining. At the beginning of training, it is initialized to $0.07$ [142]. Further, to prevent training instabilities, the temperature is clipped to avoid becoming smaller than $0.01$. Post training, the learned temperature for all CLIP models considered in this study are found to be exactly $0.01$. Moreover,

most works that fine-tune a pretrained CLIP model for different downstream tasks, use exactly this learned temperature [54, 177, 178, 197, 42, 78, 61].

Across our main experiments, we follow this standard practice of initializing $\tau$ to 0.01 and setting it to be a learnable parameter during continual pretraining. We now explore the impact of different initializations for $\tau$, and sweep over 5 different temperature values, $\{0.01, 0.1, 0.5, 0.75, 1.0\}$. From fig. 14, we observe that $\tau$ plays a crucial role for continual pretraining. As we increase the temperature from 0.01 to 0.1, zero-shot retention $\mathcal{A}_{ZS}$ gets impacted by 20% while also noting modest drops on knowledge accumulation $\mathcal{A}_{KA}$, as stability gap issues are excacerbated. Further increasing $\tau$, degrades both $A_{ZS}$ and $A_{KA}$ even more greatly, with the model degenerating to very poor performance. Such drastic changes in model behaviour were also observed in prior work investigating CLIP fine-tuning for downstream tasks [177, 97, 33]—fine-tuning at higher temperatures leads to a decrease in the modality gap between the image and text embedding spaces on the CLIP embedding hypersphere, and hence very

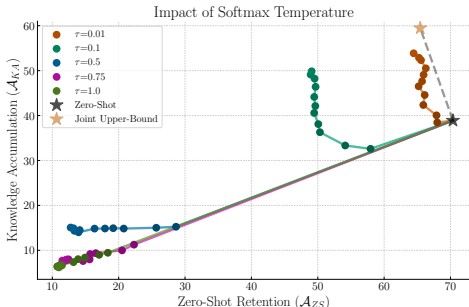

Figure 14: **The softmax temperature for the contrastive loss** is crucial for continual pretraining optimization. The learned temperature after CLIP pretraining is 0.01 (brown trajectory)—higher temperatures than the optimal 0.01 hinder continual pretraining optimization and degrade model weights.

quickly degrades the quality of the embedding space for performing downstream tasks [158, 163, 97]. We reproduce and extend the findings of these previous works for the continual pretraining regime, and emphasise the importance of retaining low temperature values for providing optimal $\mathcal{A}_{ZS}$ and $\mathcal{A}_{KA}$.

# E  Continual Pretraining: Additional Details to our Data-Centric Perspective

This section extends section 6 with detailed information on data-stream reversals specific data-pool choices and mixing ratios between streaming, buffer and pretraining data ($\mathcal{D}/\mathcal{B}/\mathcal{P}$ and $\lambda_{\mathcal{D}}, \lambda_{\mathcal{B}}, \lambda_{\mathcal{P}}$, respectively, in appendix E.2), and subsampling over the pretraining data for replay.

### E.1  What Happens if We Reverse Data-Streams?

Each sequence introduced in appendix B.4 introduces its own particular deployment scenario. Naturally, these scenarios may also either occur or be designed to occur in reverse; updating the model for example with hardest examples first, or choosing highly unrelated concepts before honing in on one specific ordering of similar concepts (by reversing `similarity`). These scenarios do not have to be related to their precursors, and can present their own unique update cycle. Evaluating fig. 6 (*right*), `random` remains consistent. The prevalent difference we find in reversing `similarity`; starting with a stream of unrelated concepts (more so than just random subsampling) and then moving towards a stream of more related concepts. Effectively, early task composition becomes forcibly harder. In doing so, the loss in retention along the trajectory comes with increased knowledge accumulation[1].

This allows the trajectory to remain consistent and close to the hypothetical linear trade-off line between the initial zero-shot behavior and the finetuning upper bound - more so even than `random` streams. Both cases however point towards high variation in the presented concepts during each update step being very beneficial for continual pretraining over longer update cycles, especially when trying to retain consistent model behaviour for each update. Still, even when also accounting for the reversed `performance` ordering, end-points converge to comparable end points! We find the only outlier to this to be the reverse `frequency` stream. As head concepts are encountered early, knowledge accumulation is lower, while the controlled placement of long-tailed, rare concepts

---

[1]By composing harder tasks, batch composition becomes also more difficult, which has been aligned with improved vision-language representation learning in *e.g.*, Zhai et al. [213]. Though by reversing `similarity` in our case, the aggregation of similar concepts towards the end of the stream results in diminished knowledge accumulation towards the end of the sequence.

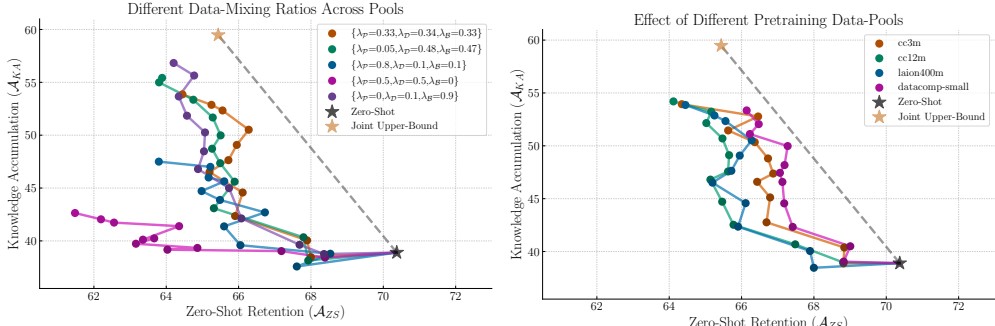

(a) **Different Data Mixture Ratios** $\lambda_{\mathcal{D}/\mathcal{P}/\mathcal{B}}$ between (b) **Quality and Diversity of the Pretraining Pool**
pretraining $\mathcal{P}$, update $\mathcal{D}$ and buffer pool $\mathcal{B}$ yield $\mathcal{P}$ can matter significantly for retention of initial zero-
significantly different adaptation-retention behaviour. shot performance.

Figure 15: **Study on Mixture Ratios and Pretraining Pools.**

towards the end of the update cycle, result in disproportionate forgetting of frequent concepts crucial
for achieving and retaining overall accumulation and retention performance.

### E.2 Data mixtures inform knowledge accumulation and zero-shot retention

Data control is also reflected in the use of different mixing ratios $\lambda_{\mathcal{P}/\mathcal{D}/\mathcal{B}}$, which we study in Fig. 15a.
The particular ratios investigated are motivated as follows (note that the baseline reference ratios we
use for all our experiments are $\{\lambda_{\mathcal{P}}{=}0.33, \lambda_{\mathcal{D}}{=}0.34, \lambda_{\mathcal{B}}{=}0.33\}$ (in orange)):

**No Buffer** $\{\lambda_{\mathcal{P}}{=}0.5, \lambda_{\mathcal{D}}{=}0.5, \lambda_{\mathcal{B}}{=}0\}$ **(in pink)** significantly degrades both accumulation and reten-
tion, hampering the $\mathcal{A}_{\text{KA}}{-}\mathcal{A}_{\text{ZS}}$ tradeoffs ($-14\%\mathcal{A}_{\text{KA}}$ and $-2.5\%\mathcal{A}_{\text{ZS}}$ compared to the reference).

**Pretrain-heavy** $\{\lambda_{\mathcal{P}}{=}0.8, \lambda_{\mathcal{D}}{=}0.1, \lambda_{\mathcal{B}}{=}0.1\}$ **(in blue)** also does not improve over the reference,
since at each update step, we input fewer update samples from $\mathcal{D}$, limiting the accumulation capacity.

**Ibrahim et al. [76]** $\{\lambda_{\mathcal{P}}{=}0.05, \lambda_{\mathcal{D}}{=}0.48, \lambda_{\mathcal{B}}{=}0.47\}$ **(in green)** defines the mixture ratio used in
past CPT work operating on LLMs. We reproduce the findings of [76], finding a $5\%$ pretraining
replay suffices to provide a better accumulation tradeoff compared to the reference ($+2.2\%\mathcal{A}_{\text{KA}}$ and
$-0.3\%\mathcal{A}_{\text{ZS}}$), suggesting that replaying pretraining data is less essential for optimal performance.

**IIDify** $\{\lambda_{\mathcal{P}}{=}0, \lambda_{\mathcal{D}}{=}0.1, \lambda_{\mathcal{B}}{=}0.9\}$ **(in violet).** Inspired by the previous result of [76], the question
arises on the importance of the overall pretraining pool $\mathcal{P}$. Extending findings in Prabhu et al. [136],
we jointly also increase the buffer mixing ratio to encourage more IID training distributions at each
update step from the full $\mathcal{D}$ and $\mathcal{B}$ pools. Doing so provides the favored tradeoff compared to all the
previous mixtures, corroborating findings in [136].

### E.3 Choice of pretraining data pool significantly impacts zero-shot retention

While the overall relevance of replay on pretraining data may be smaller than suitable buffer choices,
we complete the previous study by investigating the impact of the pretraining data pool $\mathcal{P}$ on the end
model. We experiment with three other pretraining data pools of diverse volumes, caption-sources,
curation strategies, and quality measurements—CC-3M [161], CC-12M [23], `DataComp-Small` [45]—
beyond our reference pool `LAION-400M`. For a fair comparison, we randomly subsample each
pretraining data pool to a total size of 2M samples, and use this subset as our final pretraining pool $P$.
Here too, we use the reference mixture ratio setting of $\{\lambda_{\mathcal{P}}{=}0.33, \lambda_{\mathcal{D}}{=}0.34, \lambda_{\mathcal{B}}{=}0.33\}$. From fig. 15b,
it is immediately evident that the choice of the pretraining data pool has a relevant impact on the
$\mathcal{A}_{\text{KA}}{-}\mathcal{A}_{\text{ZS}}$ tradeoffs. While adaptation capabilities are barely impacted, using `DataComp-Small` (in
pink) yields significantly better zero-shot retention properties, (upto $2.4\%\mathcal{A}_{\text{ZS}}$) gains). We speculate
that this could be attributed to the purely English-centric nature of the `CC/LAION` pools compared to
the unfiltered `DataComp-Small` which has a significantly higher multilingual and cultural diversity,
which has been shown to be beneficial for downstream performance previously [123, 124, 133].

# F   Method and Schedule Details

In the main paper, we study and reference different methods for their ability to encourage better continual multimodal pretraining on `FoMo-in-Flux`. In this section, we provide details on the methods utilized, alongside information not included in the main text with respect to the utilized learning rate schedules.

## F.1   Adaptation Methods

**LoRA [72]**   is the most commonly deployed form of parameter-efficient finetuning based on *Low-rank Adaptation*, which avoids explicitly changing pretrained weights, but instead recommends weight updates to be of the form

$$W' = W_0 + BA$$

with pretrained weights $W_0$, where $B$, $A$ are two low-rank matrices, *i.e.*, where $W \in \mathbb{R}^{d \times f}$, $A \in \mathbb{R}^{r \times f}$ and $B \in \mathbb{R}^{d \times r}$. By choosing $r << \min(d, f)$, memory requirements during finetuning can be significantly reduced. Moreover, any learned adapter weights can be absorbed into the pretraining weights. Note however that while memory is reduced, total FLOPs for backward **and** forward pass are commonly increased over simple finetuning, as full backpropagation still needs to be conducted, as noted in  Mercea et al. [118] and as consequently seen in the final MAFs breakdown (see table 4). By default, LoRA (as well as its subsequent variations VeRA and DoRA, see below) introduces an additional weighting $\alpha$ over the weight update $BA$, which we set to a constant $\alpha = 1$ [72, 88]; as it only acts as an implicit change in learning rate. As noted in Hu et al. [72], the rank $r$ is the essential hyperparameter to define for optimal changes in behaviour.

**VeRA [88]**   introduces a simple variation over LoRA by randomly initializing and freezing $A$, $B$ into fixed low-rank projections, and instead learning simple learnable vectors $\Lambda_B$ and $\Lambda_A$ such that

$$W' = W_0 + \Lambda_B B \Lambda_A A$$

where $\Lambda_B \in \mathbb{R}^f$ and $\Lambda_A \in \mathbb{R}^r$ (utilizing the same dimensional notation as above). This reduces the total number of tunable parameters significantly (though also mitigating possible adaptation capabilities), but similar to LoRA, does not positively impact FLOPs counts for backward and forward passes together.

**DoRA [104]**   minimally alters LoRA by disentangling norm and directions of the introduced adapter matrices to encourage increased stability, and moving training dynamics of LoRA-style approaches closer to those of simple finetuning. Effectively, this defines the DoRA adaptation step as

$$W' = m \cdot \frac{W_0 + BA}{\|W_0 + BA\|}$$

with magnitude vector $m \in \mathbb{R}^{1 \times f}$, where $m$ is initialized as $\|W_0\|_c$, before being jointly updated during finetuning alongside the directional (through normalization) updates induced by $B$ and $A$.

**BitFit [8]**   introduces parameter-selective model finetuning by only updating bias-terms in the model (and retaining remaining (kernel) weights as frozen). In doing so, changes to the model behaviour are supposed to be kept minimal, will still introducing several degrees of freedom for finetuning. Note however that similar to LoRA, while GPU peak memory is reduced, FLOPs are still high, as backpropagation through the full network still has to occur.

**LNFit[37]**   succeeds in the spirit of BitFit, by recommending to only tune scale and bias parameters in model architectures that leverage LayerNorm [6] layers, showcasing particular success on small continual learning benchmarks.

## F.2   Standard Continual Learning Methods

**EWC [86]**   (*Elastic Weight Consolidation*) is a regularization scheme on weight updates initially introduced to tackle rehearsal-free continual learning from scratch. The core motivation behind EWC is the assumption that for each continual task, deviation from "task-optimal" weights learned in preceding tasks should be kept meaningfully minimal. In particular,  Kirkpatrick et al. [86] argue that

deviation should be individual to each model parameter. Assuming full model weights $\theta$ after task $t$, EWC tries to approximate the curvature in parameter-loss space around $\theta_t$ via the *Fisher Information Matrix* $\mathcal{F}^t$. To estimate $\mathcal{F}^t$, several forward and backward passes have to be conducted, with the final regularization during training in task $t + 1$ defined as

$$\mathcal{L}_{t+1}^{\text{total}}(\theta) = \mathcal{L}_{t+1}(\theta) - \frac{\lambda}{2} \sum_{k \in |\theta|} \mathcal{F}_k^t (\theta_k - \theta_{t,k})^2$$

with penalty weight $\lambda$, loss function for task $t + 1$, $\mathcal{L}_{t+1}$, $\theta_t$ the weights from the previous task, and $k$ the parameter index. Note that for more than two tasks, $\mathcal{F}$ is commonly estimated through a rolling average, as done in implementation, borrowing from the `mammoth` codebase [17].

**SI [209]** (*Synaptic Intelligence*) follows a motivation conceptually related to that of EWC, in that parameters defined as more influential (by some measure) are regularized more strongly to minimize change. However, unlike EWC which computes one single point estimate using final parameter values after each task, SI computes importance measures used for regularization along the entire training trajectory. By tracking past and current parameter values, an online importance estimate is computed and incorporated as regularization as follows:

$$\mathcal{L}_{t+1}(\theta) = \mathcal{L}_{t+1}(\theta) + c \cdot \sum_{k \in |\theta|} \left( \sum_{\tau < t} \frac{\omega_k^\tau}{(\Delta_k^\tau)^2 + \zeta} \right) \left( \theta_k^t - \theta_k \right)^2 .$$

with final task weights $\theta^t$ from the previous task. Here, $\omega_k^{tau}$ is regarded as the per-parameter contribution to changes in the total loss, approximated as the running sum of the product between gradient $g_k(s) = \frac{\delta \mathcal{L}}{\delta \theta_k}$ and parameter update $\theta_k'(s) = \frac{\delta \theta_k}{\delta s}$ (with within-task update step $s$). Finally, $\Delta_k^\tau = \theta_k(s^\tau) - \theta_k(s^{\tau-1})$ estimates how much a particular parameter has moved. Alongside a simple regularization term $\zeta$ to avoid division by zero, this defines the online importance term in SI.

### F.3  Model Merging Methods

`FT-Merge` [197, 78]  introduces a simply model merging recipe, in which different finetuned variants of a same base pretrained model are linear interpolated (using interpolation coefficient $\alpha$) into a final, more general new base model. While this was initially not introduced for continual learning / pretraining tasks, this form of interpolation can be naturally extended to our problem scenario. After each task, given an interpolation coefficient $alpha$, we interpolate pre- and post-task weights ($\theta_{t-1}$ and $\theta_t$, respectively). These updated weights are then passed to the subsequent task $t + 1$. Note that we incorporate the interpolation process into the overall MAF compute budget as well.

`EMA-Merge` [172]  extends Ilharco et al. [78], but shows how a simple exponential moving average can achieve promising regularization beyond implicit learning rate changes for small, toy-ish continual learning image classification benchmarks. Similar to `FT-Merge`, `EMA-Merge` introduces an interpolation coefficient $\alpha$, and each interpolation step is account for in the overall compute budget.

`ZS-Merge`  operates in a fashion close to both merging methods - with the only differentiating factor being that after each task, interpolation occurs not with respect to preceding model weights, but instead to the initial zero-shot baseline.

## G  Differentiating Factors: `FoMo-in-Flux` with TiC-CLIP [49] and NEVIS [15]

In this section, we elaborate on the details presented in Table 1 of the main paper. We highlight the distinctive features of our benchmark, `FoMo-in-Flux`, in comparison to two closely related benchmarks: NEVIS and TiC-CLIP.

**NEVIS.** NEVIS [15], like our work, studies long-horizon continual learning with changing data distributions. However, NEVIS focuses on improving performance in a task-incremental setup, where task separation is based on dataset creation timestamps, and concentrates on performance for the current, ongoing task. In contrast, `FoMo-in-Flux` studies the ability for continual knowledge

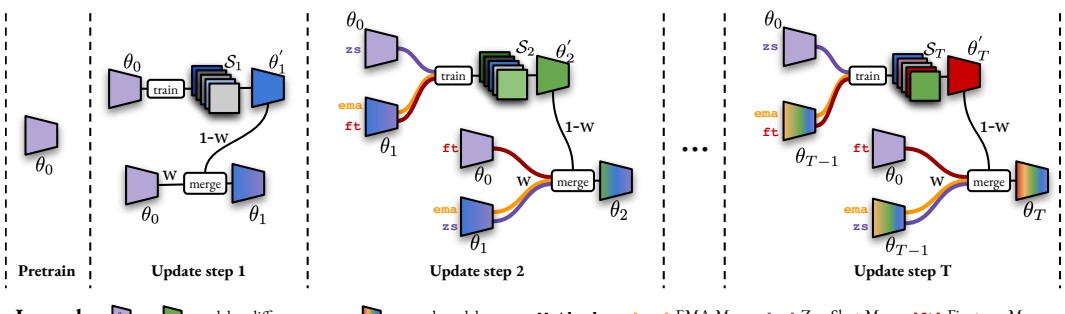

**Legend**    $\theta_0$, ..., ▉ model at different steps t    ▉ merged model     **Methods:**   [ema] EMA-Merge   [zs] ZeroShot-Merge   [ft] Finetune-Merge

Figure 16: **Different model merging strategies** explored in this work. We use $\theta'$ to denote weights $\theta$ finetuned after a respective task. Merging $\theta_{t-1}$ and $\theta'_t$ then results in the merged outputs weights for task $t$, $\theta_t$. EMA-Merge, or exponential moving average merging, merges previously merged weights $\theta_{t-1}$ with current task weights $\theta'_t$ produced by tuning the same previously merged $\theta_{t-1}$ on task $t$. ZeroShot-Merge always tunes the original pretraining weights $\theta_0$ on each task, then weight-interpolates between the finetuned $\theta'_t$ and the previously merged $\theta_{t-1}$. Finetune-Merge always interpolates between the original pretraining weights $\theta_0$ and the finetuned weights $\theta'_t$. To arrive at $\theta'_t$, the previously merged model $\theta_{t-1}$ is trained on task $t$.

*aggregation*, while balancing the retention of good downstream zero-shot performance; measuring open-ended performance in both cases and not limited to a fixed set of classes. We also tackle multimodal vision-language tasks like image-text retrieval, which are more complex to formulate than vision-only tasks. Moreover, FoMo-in-Flux allows as to study the impact of different concept and class streams to emulate task orderings that can potentially be encountered when realistically deployed.

**TiC-CLIP.** The TiC-Datacomp benchmark [49] evaluates the best methods for continual learning over *major* updates, using pretraining budgets similar to those used for pretraining CLIP. In contrast, our work focuses on *minor* updates, utilizing sample and compute scales that are $20\times-100\times$ lower than the corresponding pretraining budgets. Furthermore, TiC-CLIP operates with only six timesteps and uses large, monolithic time-incremental batches of image-text pairs. Our experiments, however, extend up to 200 timesteps and involve four carefully controlled fine-grained data-centric streams across a variety of subdomains, including medical and remote sensing images. Our study provides insights into how models can be pretrained continually over time, in scenarios working with far smaller sample and compute budgets and a larger number of timesteps, ensuring efficiency and scalability across different subdomains. Moreover, we are able to cover and study different data-centric deployment scenarios, alongside a wide array of methods and their trajectory in the knowledge aggregation and retention space. Together, FoMo-in-Flux allows us to provide the transitional benchmark towards the much more compute-intensive *major* updates as studied in TiC-Datacomp.

# H   Additional Experimental Details and Results

## H.1   Experimental Setup: Full Overview.

For complete replication, we detail the default models, compute budgets, metrics, training schedules, and data mixtures used here in significantly extended detail here.

**Pretrained Models.** We conducted our main experiments using a ViT-B-16 CLIP model pretrained on the LAION-2B dataset [159]. We also conducted some additional ablation experiments with a ViT-B-32 CLIP model (to understand the effects of different patch resolution) and ViT-S/16, ViT-L/14, ViT-H/14 and ViT-g/14 models. All our CLIP models are pretrained on LAION-2B, except for the ViT-S/16 model which is pretrained on the DataComp-1B dataset [45].

**Default Continual Pretraining Settings.** Unless otherwise specified, we always train each continual pretraining method for 20 update steps, $T=20$ (we test longer sequences with $T=\{50, 200\}$ in Supp. fig. 18). Each update step comprises of continually training a CLIP model for a fixed number

of samples derived by the computational budget outlined above. We fix the compute budgets per update step by taking the `DataComp-Small` total FLOP budget, i.e., $1.8\times10^9$ GFLOPs and dividing it by the total number of update steps. The exact number of update steps for each method is provided in Supp. Tab. 4. By default, we use a random 2M subset of `LAION-400M` as our pretraining data pool $\mathcal{P}$ and operate with uniform mixing ratios $\{\lambda_{\mathcal{P}}=0.33,\lambda_{\mathcal{D}}=0.34,\lambda_{\mathcal{B}}=0.33\}$. For our reference upper bound performance, we train a CLIP model initialized from the same `open_clip` checkpoints jointly on all 41 adaptation datasets (with the samples randomly shuffled). We do this training for a compute budget of $T \times F$ MAFs, equivalent to the overall compute budget available for the entire continual pretraining process.

**Training Details.** We train all continual pretraining methods with the CLIP contrastive loss [142, 54] and learnable temperature $\tau$, initialized to 0.01 (we provide ablations for the impact of $\tau$ initialization in appendix D.3). We select the best-reported hyperparameters for each method from previous literature, only tuning the peak learning rate for each method. We use cosine-decay LR-scheduling with linear warmup of 10% (we study more LR-schedules in section 5.1), with an AdamW optimizer [107], a batch-size of 512 [107], and clip gradients with norm higher than 1. We run all experiments using PyTorch [128]. To truly study updates in both vision and language space, we update both encoders jointly (following Zhai et al. [212], we ablate this choice in appendix D.2). Finally, the exact reflections of MAFs in method updates steps are provided in the supplementary, alongside individual reference scores finetuning CLIP on each dataset individually.

**Metrics.** From a model updating perspective, there are two main quantities of interest: the degree of *adaptation* to new data and the *retention* of pretraining knowledge. For all experiments, we therefore report two main metrics: Knowledge Accumulation ($\mathcal{A}_{KA}$), the average accuracy (or recall@5 for retrieval) over all concepts in the 41 adaptation datasets, and Zero-Shot Retention ($\mathcal{A}_{ZS}$), the zero-shot transfer accuracy (or recall@5 for retrieval) on the held-out set of 22 datasets.

**Plotting Style.** In most plots showing our main experimental result, we depict the zero-shot baseline as a black star and the joint training upper-bound as a golden star, with a dotted line connecting the two to approximate the joint training trajectory on the $\mathcal{A}_{KA}$-$\mathcal{A}_{ZS}$ plane. Every other trajectory depicts the training progression of individual experimental runs. Note that these trajectories always begin at the zero-shot baseline (black star).

## H.2 Further Replication Details.

We provide our full code, datasets, download pipeline, and experimental results here: github.com/ExplainableML/fomo_in_flux. The provided code covers all relevant details that make up `FoMo-in-Flux`: All dataset loaders, method implementations, streaming files and all generated captions for every single dataset image (c.f. `data_lib/00_info`). The code also comes with an automated downloader for preprocessed versions of each utilized dataset.

**Compute cluster and run details.** For all our experiments, we used a compute cluster with $8\times40$GB A100 nodes. For most of our ViT-B-16 runs, we used 2 GPUs from these nodes which was sufficient for all our method implementations. To ensure memory efficiency, we optimised our implementations to use CPU offloading for model weights where possible (for *e.g.*, for the `EWC`, `SI` and `Merge` methods). For comparability and reproducibility, all runs and methods share the same seed and equivalent overall experiment setting, with changes in *e.g.*, data stream ordering, modified compute budgets, method or data-mixtures only done when explicited noted.

**Justification for CLIP models used.** To ensure that our experiments were most relevant to the community, we further verified that the choice of our base CLIP models were validated by practitioner usage. On Huggingface, the `open_clip` models that were downloaded the most were CLIP ViT-B-32-laion2b (6.11M times), CLIP-ViT-H-14-laion2b (4M times), and CLIP-ViT-B-16 (2M times). Hence, we investigate these models - particularly as ViT-B/16 has been used in other studies on continual major model updates such as Garg et al. [49].

**Exact number of update steps, MAFs and samples seen.** We provide the full breakdown of how we compute MAFs per time step for each of the methods, and the total compute budget in terms of samples seen per method (in Appendix table 4). We use the `datacomp-small` [45] compute budgets as our reference. Hence, this means that our total compute budget for the full continual

Table 4: **Compute Budgets used in all ViT-B-16 experiments.** We provide the total number of GFlops taken per task for each of the methods in the `Per-Task GFlops` column. We also showcase the maximum GPU memory requirements for each method in the `Max. Memory Reqd.` column— we convert this into a memory multiplier for each method by dividing with respect to the reference `full-ft` max memory required. Finally, for each method the `Per-Task MAFs` are computed as the product of the `Per-Task GFlops` and the `Memory Multiplier`. Then, we show the total number of gradient update steps that are allowed for these compute budgets per update step $t$, for the four total number of time step settings, $T=\{20,50,100,200\}$. Finally, we also show the total number of gradient steps used (`Total Num. steps`) and the total number of samples seen (`Total Num. samples seen`) for the full continual pretraining process—our joint upper-bound oracle also uses this total compute budget.

| Method | Per-Task GFlops | Max Memory Reqd. | Memory Multiplier (wrt full-ft) | Per-Task MAFs | Num. steps ($T$=20) | Num. steps ($T$=50) | Num. steps ($T$=100) | Num. steps ($T$=200) | Total Num. steps | Total Num. samples seen |
|---|---|---|---|---|---|---|---|---|---|---|
| full-ft | 63394.7585 | 46.5917 | 1 | 63394.7585 | 1420 | 568 | 284 | 142 | 28,400 | 14,540,800 |
| locked-text | 57254.6183 | 37.5761 | 0.8064 | 46170.1241 | 1949 | 780 | 390 | 195 | 39,000 | 19,968,000 |
| locked-image | 27176.6698 | 11.8847 | 0.2551 | 6932.7684 | 12982 | 5193 | 2596 | 1298 | 259,600 | 132,915,200 |
| LNFit | 43165.5968 | 30.5566 | 0.6558 | 28307.9983 | 3179 | 1272 | 636 | 318 | 63,600 | 32,563,200 |
| BitFit | 43165.5968 | 30.5546 | 0.6558 | 28307.9983 | 3179 | 1272 | 636 | 318 | 63,600 | 32,563,200 |
| LoRA, r=4 | 54479.2515 | 40.5449 | 0.8702 | 47407.8446 | 1898 | 759 | 380 | 190 | 38,000 | 19,456,000 |
| LoRA, r=64 | 54505.0151 | 40.6757 | 0.873 | 47582.8781 | 1891 | 757 | 378 | 189 | 37,800 | 19,353,600 |
| DoRA, r=4 | 54479.8241 | 40.6582 | 0.8726 | 47539.0945 | 1893 | 757 | 379 | 189 | 37,800 | 19,353,600 |
| DoRA, r=64 | 54514.1754 | 40.7871 | 0.8754 | 47721.7091 | 1886 | 754 | 377 | 189 | 37,800 | 19,353,600 |
| VeRA, r=4 | 54479.3393 | 40.5449 | 0.8702 | 47407.921 | 1898 | 759 | 380 | 190 | 38,000 | 19,456,000 |
| VeRA, r=64 | 54507.8336 | 40.5742 | 0.8708 | 47465.4214 | 1896 | 758 | 379 | 190 | 38,000 | 19,456,000 |
| EWC | 6276081.094 | 47.207 | 1.0132 | 6358925.364 | 14 | 6 | 3 | 1 | 200 | 102,400 |
| SI | 63394.7585 | 46.6523 | 1.0013 | 63477.1716 | 1418 | 567 | 284 | 142 | 28,400 | 14,540,800 |
| ZS-Merge | 63394.7585 | 46.5917 | 1 | 63394.7585 | 1420 | 568 | 284 | 142 | 28,400 | 14,540,800 |
| FT-Merge | 63394.7585 | 46.5917 | 1 | 63394.7585 | 1420 | 568 | 284 | 142 | 28,400 | 14,540,800 |
| EMA-Merge | 63394.7585 | 46.5917 | 1 | 63394.7585 | 1420 | 568 | 284 | 142 | 28,400 | 14,540,800 |

pretraining is set to $5.7{\times}10^8$ GFlops for the ViT-B-32 architecture and $1.8{\times}10^9$ GFlops for the ViT-B-16 architecture.[2]

**Variance across seeds.** To ensure that our results are statistically valid and generalizable, we re-run our canonical continual pretraining experiment with a ViT-B/16 backbone on the 20-task random data stream, with three different seeds. fig. 17 showcases that the three trajectories across the different seeds result in very similar patterns and low variance across runs. This validates that all our main results are generalizable across seeds.

**Additional Experiment Results.** Finally, we augment our suite of experiments conducted in the main paper.

Fig. 18 provides additional higher-level experiment insights and verification, covering changes in backbone architecture, compute budget and total update steps / task counts. More precisely, Fig. 18 (*left*) shows the impact an increase or decrease in overall compute budget has. As can be seen, all trajectories behave similarly on a qualitative level - experiencing forgetting and stability gap [36] issues at the beginning, before recovering towards the linear zeroshot-finetuning trend line. Comparing end points, we do find that larger compute budgets encourage slightly increased knowledge accumulation gains, but at the cost of disproportionately larger losses in knowledge retention. This means that in practice, large compute budgets may be less favoured even from a performance standpoint to incorporate minor model updates and bridge time between large, major model updates. On top of that, Fig. 18 (*right*) highlights that under a fixed compute budget, in order to bridge time to large model updates, keeping the number of minor model updates small, while maximizing the size of each respective minor update, is preferable from both a knowledge accumulation and retention perspective. Further, we note the strong robustness of model merging even under very long task streams, further strengthening their applicability for long-step continual pretraining.

Fig. 18 (*center*) augments our results on the impact of different data-centric deployment scenarios for continual minor model updates, under a different patch resolution for the vision-transformer. In this experiment, we continually pretrain ViT-B-32 image-encoder models instead of the standard ViT-B-16 image-encoder. We note that the overall trends from this experiment closely match those of

---

[2]Note that the compute budgets outlined in the original paper [45] were in GMacs—we convert these numbers to GFlops by multiplying by 2 (see here for reference.)

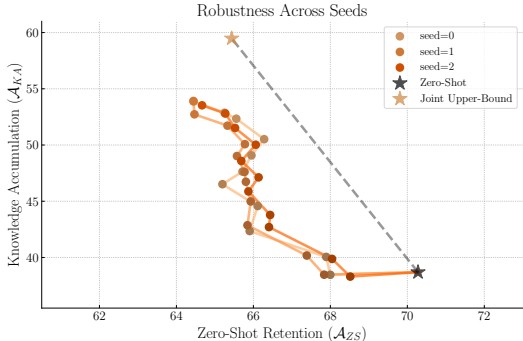

Figure 17: Our continual pretraining insights are robust across different random seeds—the variance in trajectories across three different seeds is minimal.

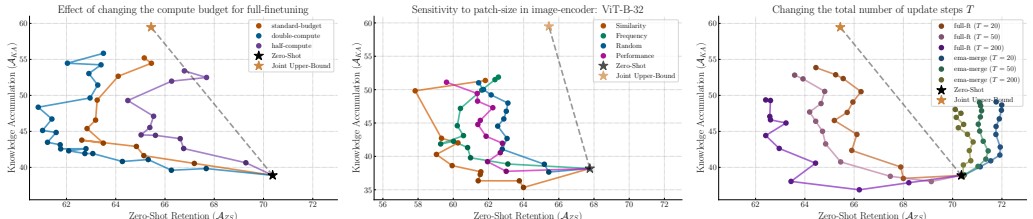

Figure 18: We provide additional experiment insights and verifications, covering changes in backbone architecture, compute budget and update steps. *(Left)* Changing the available compute budget noticeably affects knowledge retention, however with limited gains in knowledge accumulation. *(Center)* Replacing our default patch-size of $16 \times 16$ to $32 \times 32$ (*i.e.*, ViT-B-16 to ViT-B-32) for ablating the effect of lower the patch-resolution of our vision-transformer backbones, retains comparable behaviour across different deployment scenarios, with surprisingly similar trajectory endpoints, and comparable accumulation performance. *(Right)* Changing the number of tasks the data stream (referred to as update steps $T$) is divided into, we find drops in both knowledge retention and accumulation. Correspondingly, these results generally recommend to keep the number of minor updates as small as possible, and the respective sizes as large as can be. Note that each trajectory has been uniformly subsampled to visualize the same number of trajectory points for better visual readability. Additionally, note that the robustness of the `EMA-Merge` method extends to longer task streams, reinforcing its potential as a strong approach for continual pretraining.

the original ViT-B-16 experiments ( fig. 6), suggesting the overall robustness of our main data-centric results to the patch-resolution of the input images.

# I `FoMo-in-Flux`: Datasets

## I.1 Finetuning verification

In order to estimate a reference upper bound on adaptation performance, we verify the quality of generated captions, and perform a sanity-check on our training pipeline, we fine-tune ViT-B/32 and ViT-B/16 individually on the datasets in our training split, as well as the evaluation-only datasets which come with training samples. We fine-tune the model on each dataset for 10 epochs with the same learning rate scheduling and the results are shown in table table 5. As can be seen, we find a consistent, and in parts significant improvements conducting CLIP-style training across all individual benchmarks—highlighting the validity of our generated captions, and support for each benchmark to be included in `FoMo-in-Flux`.

Table 5: Per-dataset fine-tuning results for the ViT-B/32 and ViT-B/16 backbone. FT Performance is the maximum accuracy over 10 epochs. Delta to ZS is the difference between FT Performance and the initial zero-shot accuracy.

| Dataset | ViT-B-16 | | ViT-B-32 | |
| --- | --- | --- | --- | --- |
| | FT Performance | Delta to ZS | FT Performance | Delta to ZS |
| Ai2Diagrams [84] | 88.00 | 10.67 | 83.67 | 12.33 |
| ArtBench10 [99] | 22.86 | 11.64 | 21.20 | 9.08 |
| Birdsnap [9] | 63.70 | 13.30 | 57.60 | 10.00 |
| Caltech101 [94] | 93.33 | 1.33 | 93.67 | 1.67 |
| Caltech256 [55] | 93.97 | 1.39 | 92.61 | 2.61 |
| Cars196 [169] | 93.88 | 5.07 | 90.56 | 2.25 |
| Cifar100 [93] | 90.33 | 15.83 | 91.33 | 15.93 |
| Cifar10 [91] | 99.67 | 4.67 | 99.00 | 4.70 |
| CLEVR [83] | 71.05 | 67.19 | 55.87 | 52.94 |
| CLRS [151] | 92.67 | 29.33 | 91.33 | 30.00 |
| Country211 [142] | 20.38 | 3.74 | 20.38 | 6.11 |
| CUB200 [186] | 80.50 | 10.38 | 74.00 | 10.27 |
| DF20mini [131] | 50.84 | 49.46 | 43.30 | 41.64 |
| DollarStreet [152] | 18.31 | 11.88 | 17.96 | 12.26 |
| DomainNet-Clipart [129] | 83.62 | 3.14 | 81.74 | 3.93 |
| DomainNet-Infograph [129] | 61.16 | 3.71 | 54.93 | 2.55 |
| DomainNet-Painting [129] | 74.64 | 3.61 | 71.72 | 1.47 |
| DomainNet-Quickdraw [129] | 66.81 | 48.45 | 66.52 | 48.24 |
| DomainNet-Sketch [129] | 78.26 | 3.94 | 76.96 | 4.89 |
| Dsprites [115] | 100.00 | 88.16 | 100.00 | 88.36 |
| DTD [31] | 68.00 | 16.00 | 66.33 | 11.33 |
| EuroSAT [65] | 99.67 | 43.62 | 99.33 | 47.85 |
| FashionMNIST [201] | 96.33 | 16.93 | 94.67 | 18.07 |
| FGVCAircraft [110] | 48.67 | 22.24 | 39.33 | 14.41 |
| Flowers102 [125] | 95.67 | 21.33 | 94.67 | 21.33 |
| Food101 [18] | 90.67 | 5.08 | 88.00 | 5.66 |
| FRU92 [69] | 91.67 | 42.97 | 88.33 | 39.64 |
| GTSRB [71] | 99.33 | 49.46 | 100.00 | 56.12 |
| iNaturalist2021 [79] | 50.40 | 44.76 | 43.10 | 37.80 |
| Isicmelanoma [41] | 59.33 | 51.00 | 56.00 | 40.33 |
| MITStates [80] | 28.30 | 4.75 | 26.35 | 3.02 |
| MNIST [40] | 100.00 | 34.70 | 99.67 | 30.57 |
| Monkeys10 [2] | 97.79 | 15.07 | 96.69 | 13.97 |
| MTSD [44] | 90.97 | 72.41 | 90.75 | 70.93 |
| MVTec-AD (Base) [10] | 100.00 | 27.67 | 100.00 | 21.00 |
| MVTec-AD (Faults) [10] | 52.33 | 38.67 | 38.00 | 20.67 |
| ObjectNet [7] | 54.63 | 16.75 | 48.88 | 16.98 |
| Obscure Animals | 89.67 | 27.49 | 89.33 | 33.78 |
| Obscure Things | 73.33 | 17.54 | 68.67 | 14.98 |
| OpenImages [90] | 58.64 | 0.00 | 59.40 | 0.38 |
| OxfordPets [126] | 95.00 | 4.29 | 90.67 | 0.23 |
| PatternNet [226] | 99.67 | 30.72 | 99.67 | 34.14 |
| Places365 [221] | 48.49 | 6.62 | 49.86 | 7.22 |
| Plantvillage [75] | 100.00 | 80.02 | 99.67 | 76.55 |
| Quilt-1M [77] | 66.45 | 65.45 | 67.10 | 66.80 |
| Resisc45 [68] | 94.33 | 25.60 | 93.33 | 30.16 |
| Shapes3d [68] | 100.00 | 87.16 | 100.00 | 85.68 |
| SnakeCLEF2023 [130] | 22.17 | 21.98 | 16.51 | 16.45 |
| SUN397 [202] | 75.69 | 6.22 | 73.93 | 5.62 |
| STL10 [32] | 100.00 | 3.25 | 98.67 | 1.42 |
| SVHN [122] | 99.33 | 46.32 | 99.00 | 57.01 |
| SynthClip106 [60] | 46.67 | 5.46 | 44.00 | 4.30 |
| VEG200 [69] | 84.75 | 53.90 | 79.50 | 46.70 |
| Zappos50k [205] | 35.14 | 22.36 | 31.29 | 18.25 |

## J  `FoMo-in-Flux`: Caption Pipeline

As part of our `FoMo-in-Flux` pipeline, we converted 63 different classification and retrieval datasets into a format that made them amenable for contrastive language-image pretraining. This entailed providing text captions for each of the images in the classification datasets. For this, our main aims were to ensure: (1) *scalability* of the captioning pipeline, (2) that the captions captured *real-world and fine-grained* details about the image, (3) that the captions were *not verbose* so that they would fit into the context length of CLIP's text encoder (77 tokens), and (4) that the captions *contained the true classname* of each of the images from the classification datasets.

To this end, we proceeded to caption the images in a three-stage manner—(1) We first used a BLIP-2 model [96] using a T5-XL decoder to ensure high captioning performance along with scalability to provide initial seed synthetic captions for each of the images, (2) we next generated templated captions for each of the images using the classnames, for *e.g.*, for an image of a `tench` in the ImageNet dataset, we use a templated caption, "A photo of a tench" and similarly for an image of a `manted howler` in the Monkeys10 dataset, we use a templated caption, "A photo of a mantled howler, a type of monkey.", and finally (3) we merge both the templated and seed synthetic captions using the Capsfusion [207] model—a LLaMA model that is finetuned to take in two captions for an image, and return a merged caption capturing the key aspects of both the captions. Using our three-stage pipeline, we are able to generate diverse yet faithful captions for each of the images in our set of 63 datasets. We showcase a visualisation of our generated captions for some of our constituent datasets in fig. 19.

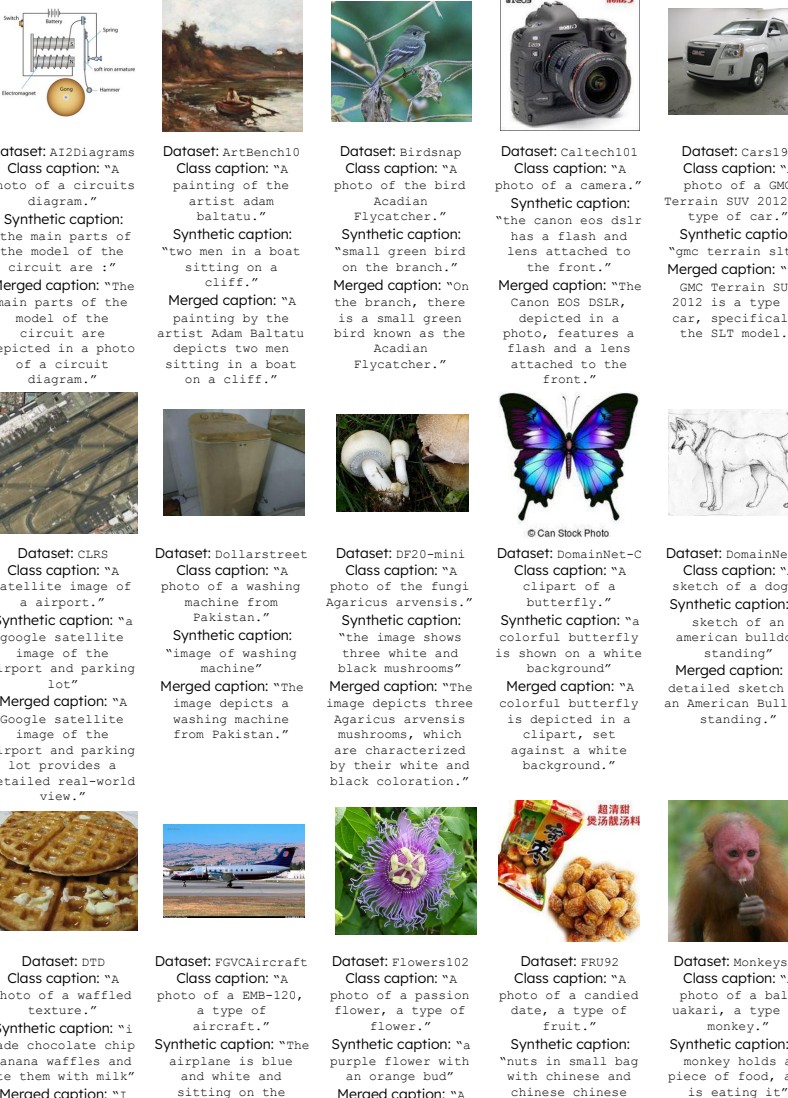

Figure 19: **Random Samples from** `FoMo-In-Flux`. We showcase some sample captions generated using our three-stage pipeline for a few of the datasets in `FoMo-In-Flux`. The `Class caption` is the templated caption using the class-name, `Synthetic caption` is the caption generated using BLIP-2, and the `Merged caption` is the final merged caption using Capsfusion (merging both `Class caption` and `Synthetic caption`).

# K  Data Statement

Dataset Title: `FoMo-in-Flux`

Dataset Curator(s): N/A

Dataset Version: 1.0

Dataset Citation: N/A

Data Statement Authors: N/A

Data Statement Version: 1.0

Data Statement Citation and DOI: N/A

## K.1  Executive Summary

`FoMo-in-Flux` is an aggregate benchmark comprising over 2.53M images from 63 classification and retrieval datasets, including 61 existing datasets and 2 newly introduced ones, described in appendix B. On top of image and labels provided by the original datasets, we provide a caption for each image, generated using the pipeline described in appendix J.

## K.2  Curation Rationale

`Fomo-in-Flux` is a benchmark for continual multimodal pretraining that emphasizes adaptation across distinct subdomains over long time horizons, while allowing for finegrained controllability of particular concepts and classes presented at respective update steps for a data-centric perspective on continual multimodal pretraining. The constituent datasets were selected based on availability, licensing, quality of labels, diversity of data domains, quality of the resulting captions, and the degree of adoption in the computer vision and machine learning research communities.

## K.3  Documentation for Source Datasets

The licensing information for source datasets, as well as relevant citations, are provided in table 2 and table 3. We release the captions, as well as the Obscure Animals and Obscure Things datasets under the MIT license (https://opensource.org/license/mit).

## K.4  Language Varieties

All the class labels and captions are in English.

## K.5  Speaker Demographic

N/A

## K.6  Annotator Demographic

The captions were created using an automated pipeline and based on original class labels, as outlined in appendix J. For selected simpler datasets, we use the templated captions directly, as shown in table 2 and table 3. For the information about annotators of source datasets, please see the references in table 2 and table 3.

## K.7  Speech Situation and Text Characteristics

N/A

## K.8  Preprocessing and Data Formatting

The class labels are used as-is with no modification. All images are resized to 224x224 pixels.

### K.9  Capture Quality

N/A

### K.10  Limitations

Although great care was taken to ensure the correctness of the dataset and random samples of the captions were manually inspected for a quality check, we did not verify the captions for all 2.53M samples. Given the dependence on BLIP-2 [96] and Capsfusion [207], the captions might reflect the biases and idiosyncracies of these models.

Moreover, as an aggregate benchmark, `Fomo-in-Flux` reflects the data collection and annotation biases of the source datasets. However, by pooling diverse sources of data, we avoid a systematic dataset-wide curation bias.

### K.11  Broad Impact

Our dataset helps assess the continual multimodal pretraining performance across various methods, data stream orderings, learning rate schedulers, and compute budgets. The insights gained will help optimize continual pretraining, facilitating fewer large-scale model updates. This optimization, in turn, will help decrease energy consumption and lower carbon emissions associated with continual adaptation of foundation models, and overall encourage a more economical and ecological treatment of these large architectures.

### K.12  Metadata

License: https://opensource.org/license/mit

Annotation Guidelines: N/A

Annotation Process: Automatic

Dataset Quality Metrics: N/A

Errata: N/A

### K.13  Disclosures and Ethical Review

N/A

### K.14  Other

N/A

### K.15  Glossary

N/A

**About this data statement**

A data statement is a characterization of a dataset that provides context to allow developers and users to better understand how experimental results might generalize, how software might be appropriately deployed, and what biases might be reflected in systems built on the software.

This data statement was written based on the template for the Data Statements Version 2 Schema. The template was prepared by Angelina McMillan-Major, Emily M. Bender, and Batya Friedman and can be found at http://techpolicylab.uw.edu/data-statements.

