# OpenReview forum: "A Practitioner's Guide to Real-World Continual Multimodal Pretraining"
_NeurIPS.cc/2024/Datasets_and_Benchmarks_Track — NeurIPS 2024 Track Datasets and Benchmarks Poster_

### Official Review · Reviewer_Fxhw · 2024-07-14
**The paper provides a comprehensive benchmark for continual multimodal pretraining from multiple perspective providing a useful resource to the community.**

**Rating:** 7
**Confidence:** 4
**Correctness:** the evaluation methods and experiment…
**Clarity:** The paper is well organized and easy …

**Review:**

The paper provides a comprehensive benchmark for continual multimodal pretraining.
The paper is well organized and the experiments seem well designed.
The benchmark could be a valuable resource for practitioners.

**Strengths:**

1. Paper is well organized and easy to follow
2. Experiments seem well designed and comprehensive

**Additional Feedback:**

NA

**Documentation:**

sufficient details are provided in the main paper and appendix.

**Ethics:**

I do not suspect any particular ethical concerns in this papre.

**Opportunities For Improvement:**

Why does the paper sometimes use the word 'pretraining' and other times use the word 'finetuning'? Are they considered the same in the context of this paper?

**Relation To Prior Work:**

Relation to Prior Work is discussed (Table1)

**Summary And Contributions:**

The paper provides a comprehensive benchmark for continual multimodal pretraining. The benchmark covers 63 datasets. The experiments seem well designed and valuable insights are obtained.

---

> ### Author Rebuttal · Authors · 2024-08-17
>
> We would like to sincerely thank the reviewer for their positive feedback and thoughtful comments. We are glad the reviewer recognised our **"comprehensive benchmark"** and **"well-designed experiments"** as a **"valuable resource for practitioners"**. Lastly, we are grateful for their assessment of our work as **"well-organised"** and **"easy to follow"**. We seek to address the reviewer’s concerns and questions below:
>
> > Why does the paper sometimes use the word 'pretraining' and other times use the word 'finetuning'? Are they considered the same in the context of this paper?
>
> Thank you for your feedback and for pointing out the confusion. To clarify, **'continual pretraining'** refers to our overarching goal of updating foundation models over time with new data.
>
> In our approach, **'finetuning'** is a baseline continual learning method, where finetuning means simply continuously training with the standard CLIP contrastive loss, without including additional measures to prevent catastrophic forgetting. We will better clarify this in the final draft.
>
> > The benchmark could be a valuable resource for practitioners.
>
> To even further improve the utility of our benchmark, we have added a comprehensive list of additional experiments, and believe that it increases the value and generality of insights provided in our work even further. We detail the experiments performed in the common response.
>
> Overall, thank you for your encouraging review and positive comments.

---

> > ### Comment · Reviewer_Fxhw · 2024-08-18
> >
> > I have read the author's response and the comments from other reviewers. I think this is could be a valuable resource to practitioners and would like to maintain my rating of the paper.

---

### Official Review · Reviewer_jRa8 · 2024-07-23

**Rating:** 8
**Confidence:** 4
**Clarity:** The paper is well written.

**Review:**

**Quality**: The paper is well-researched, making it a high-quality contribution to the field.

**Clarity**: The writing is clear and the explanations are easy to follow.

**Originality**: The introduction of the FoMo-in-Flux benchmark is a novel contribution.

**Significance**: This work is significant as it provides practical guidelines and insights that can help practitioners improve their CPT tasks.

**Strengths:**

1. **Introduction of a Novel Benchmark (FoMo-in-Flux):** The paper introduces FoMo-in-Flux, a new benchmark specifically designed for continual pretraining (CPT). This benchmark includes 63 diverse datasets and allows for fine-grained control over data streams and compute budgets. This innovation fills a gap in the current literature, providing a robust framework for testing and comparing different CPT methods under realistic conditions.

2. **Detailed Experiments Providing Practical Insights:** The authors conducted extensive experiments on the FoMo-in-Flux benchmark, exploring various deployment scenarios, data mixtures, and pretraining data pools. These experiments offer practical insights into how different factors affect knowledge accumulation and zero-shot retention. The findings help practitioners make informed decisions on optimizing their CPT processes, making the research highly applicable to real-world tasks.

3. **Clear and Easy-to-Understand Explanations:** Despite the complexity of the subject, the paper is written in a clear and straightforward manner.

4. **Evaluation of Various Training Strategies:** The paper evaluates a range of continual learning and finetuning strategies, such as parameter-efficient methods like LoRA and model merging techniques like EMA-merge. By comparing these methods, the authors provide valuable guidance on selecting the most effective strategies for balancing knowledge retention and adaptability in CPT tasks. This comprehensive evaluation adds depth to the research and aids practitioners in choosing the best approaches for their needs.

**Additional Feedback:**

No

**Correctness:**

The claims made in the paper appear to be correct based on the thorough and methodical approach used in the research.

**Documentation:**

The authors provide the code for the benchmark and used datasets.

**Limitations:**

As shown in above.

**Opportunities For Improvement:**

1. **Need for Additional Experimentation for Practical Applications:** Although the paper provides extensive experimental results, some practical applications might require further experimentation beyond what is presented. Practitioners may need to conduct additional tests to tailor the insights to their specific use cases or to explore scenarios not covered in the paper. This suggests that while the research is a strong starting point, it may not cover all practical aspects comprehensively.

2. **Missing Comparative Analysis**: The paper does not provide a comparison with existing benchmarks or papers. The authors do not provide a detailed comparison that highlights the unique aspects and novel contributions of the FoMo-in-Flux benchmark relative to previous works in the field.

**Relation To Prior Work:**

The paper does not have a dedicated section that explicitly discusses how this work differs from previous contributions.

**Summary And Contributions:**

The paper introduces the FoMo-in-Flux benchmark, designed to study continual multimodal pretraining under realistic constraints. It provides detailed insights into the impact of deployment scenarios, data mixtures, and pretraining data pools on knowledge accumulation and zero-shot retention. The study evaluates various continual learning and finetuning strategies, highlighting the benefits of parameter-efficient methods like LoRA and model merging techniques like EMA-merge. It also explores different learning rate schedules, emphasizing the effectiveness of autoregressive meta-schedules derived from infinite learning rate variants. The paper offers practical guidelines for practitioners, covering deployment structures, data mixtures, method choices, and learning rate schedules to facilitate the effective implementation of continual pretraining in real-world applications.

---

> ### Author Rebuttal · Authors · 2024-08-17
>
> We would like to sincerely thank the reviewer for their positive feedback and thoughtful comments. We are glad that they found our work ***well-researched***, ***a high-quality contribution***, and ***a significant contribution*** and appreciated our ***extensive experiments*** and the applicability of our research to ***real-world tasks***. We are glad to hear that our work is ***clear and easy to understand***. We address the reviewer’s concerns and questions below:
>
>
> > Need for Additional Experimentation
>
>
> We agree with the reviewers feedback that additional experimentation would increase the impact of this work even further. As such, we have added a number of additional experiments to address this concern and make our study even more comprehensive (see also our shared reponse):
>
> 1. ***Model Scaling***:
> We provide additional experimental examination of the effect of scaling model sizes for continual pretrainability across longer update horizons here: https://bit.ly/4fRf7H6.
>
> Our findings reveal that increasing model size significantly improves the retention of generalization performance without impacting knowledge accumulation rates. As a result, we find that larger models not only acquire substantial new knowledge with reduced forgetting but also occasionally demonstrate even positive backward transfer, which is quite remarkable. From a long-term deployment perspective, larger models prove to be more adaptable for future updates.
>
> 2. ***Compute Scaling***:
>
> We also explored how varying compute budgets impact performance for a given model size, providing a detailed analysis of efficiency concerning compute resources. The results are available here: https://bit.ly/4ctYSgg.
> We observed that simply increasing the compute budget for a fixed model size does not inherently lead to a more favorable accumulation-versus-retention trade-off during fine-tuning. However, when combined with model merging, larger compute budgets significantly enhance the accumulation and retention trade-off, showing a near log-linear relationship between compute resources and the accumulation-retention trade-off.
>
>
> 3. ***Additional Data-centric Insights:***
>
> 3.1 ***Dataset-Incremental and Time-Incremental Streams:***
> To extend the applicability of our results to even more data-centric deployment scenarios, we incorporated two additional dataset-level streams: random dataset ordering for task grouping and a NEVIS-inspired time-incremental dataset ordering. In both scenarios, we observed similar behavior, characterized by high trajectory variance due to the anticipated high visual biases within each dataset (refer to Torralba et al. 2011 “Unbiased look at dataset bias”, Liu et al. 2024 “A Decade’s Battle on Dataset Bias”). The results are visualized here: https://bit.ly/4cpvNmg.
>
> 3.2 ***Reversed Data Streams***:
> Additionally, we included reversed versions of each stream, which themselves present unique scenarios. For instance, reversing the “similarity” stream leads to highly dissimilar concepts in the initial updates, with a convergence of concepts towards the end. This reversal notably impacts the trajectory, resulting in more consistent performance than even randomized streaming (https://bit.ly/3SUQnE1). Importantly, despite the addition of these data streams, we find further evidence that update cycles over the same data distribution tend to converge towards similar performance endpoints.
>
> 4. ***New Methods (GaLore)***:
> We integrated GaLore (Zhao et al.) for gradient-based rank approximations as an additional conceptual approach to continual pretraining. This method presents an intriguing accumulation-retention tradeoff, offering a balanced middle ground between fine-tuning and parameter-efficient fine-tuning. More details can be found here: https://bit.ly/3WN6a93.
>
>
>
> These updates provide an even more thorough evaluation of our methods and offer deeper insights into their behaviour under different conditions. We believe these additions strengthen our contribution to the field and we hope they sufficiently address the reviewer’s concerns.

---

> > ### Author Rebuttal · Authors · 2024-08-17
> >
> > > Missing Comparative Analysis:
> >
> >
> > We create a controlled, large scale benchmark that for the first time allows controlled continual multimodal pretraining of CLIP models by adding fine-grained contextual image captions to 63 datasets and enable their use for CLIP pretraining. To separate FoMo-in-Flux from even loosely related works, we have included a comprehensive tabular comparison in Table 1 in the main paper (which has now also been extended with additional separators, see shared response).
> >
> > However, we agree that our discussion of related continual learning and continual pretraining works is not as explicit as it could, as it has been integrated into our new perspective on continual pretraining in Section 2.1. We have thus moved this part into its own section, and clearly highlighted that this section discusses related works.
> >
> > Moreover, we kindly point the reviewer to Appendix Section 3, where we also provide an extended comparison with the two closest benchmarks (as beyond the pure benchmark aspect, our comprehensive continual pretraining study and perspective on it is inherently novel), provided in summary here for reference:
> >
> > ***NEVIS***: NEVIS focuses on image-only, task-incremental continual learning tasks with changing data distributions and evaluates performance based on current tasks, which are distinguished by dataset creation timestamps. Most experiments are small models trained from scratch. In contrast, FoMo-in-Flux examines continual knowledge aggregation in foundational multimodal models with a focus on maintaining robust zero-shot performance across an open-ended set of classes. Moreover, FoMo-in-Flux allows for complex, concept-controlled continual multimodal/vision-language pretraining; which is fundamentally impossible in NEVIS.
> >
> > ***TiC-CLIP***: Within our continual pretraining perspective, TiC-CLIP occupies the major update endpoint, and evaluates continual learning methods over vast updates using extensive pretraining budgets, with a focus on six timesteps and large batches of image-text pairs. Our approach, FoMo-in-Flux, deals with minor updates and operates on smaller, but in turn controlled sample and compute scales. It extends over a much longer update horizon, and involves fine-grained controllable data streams from diverse subdomains - something not possible to a comparable degree in TiC-CLIP. This allows us to study several new perspectives such as data-centric deployment scenarios, a much more extensive study of update strategies, and the impact of e.g. meta learning rate schedules. Together, FoMo-in-Flux offers a unique, scalable and efficient framework for continual learning in the longer horizon, controlled minor update range, bridging the gap to more compute-intensive major updates as seen in TiC-CLIP.

---

### Official Review · Reviewer_4Xf5 · 2024-07-25
**Foundation modal pretraining with diverse large dataset**

**Rating:** 7
**Confidence:** 2
**Correctness:** The claims seems to be correct.

**Review:**

+ Combination of large diverse dataset
+ Insights on setup and parameter effects

- No new dataset introduced
- Dataset combination limit the potential multimodal application.

The overall quality of the paper is good and clear to follow. While this paper seems to be original, it lacks significant difference if compared to some other existing work, especially when the dataset are pre-existing. The insights provided with pretraining seems to be limited and may lack a broader audience due to the choice of the model and diversity of the benchmark. This may reduce the significance of this work but overall provide good reference for other researchers and users.

**Strengths:**

The strength mainly comes from the insights on the setup and parameters. It also provides an idea for other researchers to construct their own benchmark for intended use case. The reference is also extensive and provides good entry point for new researchers. The writing is easy to follow.

**Additional Feedback:**

None.

**Clarity:**

The paper is in good quality, although the fonts of the figures can be improved.

**Documentation:**

The documentation seems to be adequate, and can be reproduced.

**Ethics:**

None.

**Limitations:**

The authors didn't mention any limitations and potential negative societal impact of their work. However, the general limitation and impact are limited based on this work alone.

**Opportunities For Improvement:**

Multiple models can provides a much more convincing results on how the insights will have a broader audience and can be applied to different applications.

**Relation To Prior Work:**

The relation is clearly discussed.

**Summary And Contributions:**

This paper propose an infusion of multiple dataset to create a benchmark for multimodal foundation model pretraining. The main contributions are the combination of the dataset and insights on how pretraining on the dataset looks like with different setup and parameters.

---

> ### Author Rebuttal · Authors · 2024-08-17
>
> We would like to sincerely thank the reviewer for their positive feedback and thoughtful comments. We are grateful for the recognition of our work as **good quality** and **clear to follow**, as well as for acknowledging its **originality** and value as a **good reference for other researchers**. We also appreciate the praise for our **extensive references** and for providing **ideas for future research**. We seek to address the reviewer’s concerns and questions below:
>
> > Multiple models can provide much more convincing results on how the insights will have a broader audience and can be applied to different applications.
>
> We agree with the reviewer that a broader range of results will make our work even more impactful. As per our shared response, we have added the following experiments:
>
> 1. ***Model Scaling:*** We conducted experiments to evaluate the impact of scaling model sizes on performance, providing insights into how different model capacities affect longer horizon continual pretrainability. Our results are shown here: https://bit.ly/4fRf7H6
>
> We discover that when increasing the model size, retention of generalisation performance is much less at odds with knowledge accumulation. Increased capacity helps the model acquire substantial new knowledge while minimising forgetting, and even in parts allows for positive backward transfer, which is quite surprising! From a pure long-term deployment perspective, we find that larger models offer better updatability down the line.
>
> 2. ***Compute Scaling:*** We also investigated how changing the compute budget influences performance for a given model size, offering a detailed analysis of efficiency in relation to compute resources. Results are visualised here: https://bit.ly/4ctYSgg
>
> We discover that for a fixed model size, increasing the compute budget does not automatically come with a more favourable accumulation-versus-retention trade-off when simply fine-tuning. However, in conjunction with model merging, larger compute budgets provide a much better,  near log-linear relation between compute and the accumulation-retention tradeoff.
>
> 3. ***Additional Data-centric insights:***
>
> 3.1 *Dataset-Incremental and Time-Incremental Streams:* Moreover, to enable transfer of our results to even more data-centric deployment scenarios, we have included two additional dataset-level streams: task grouping based on random dataset ordering, and a NEVIS-inspired time-incremental dataset ordering. In both cases, we find comparable behaviour, namely high trajectory variance attributable to the expected high visual biases within each dataset (c.f. Torralba et al. 2011 “Unbiased look at dataset bias”, Liu et al. 2024 “A Decade’s Battle on Dataset Bias”). Results are visualised in https://bit.ly/4cpvNmg.
>
> 3.2. Reversed Data Streams: Moreover, we incorporate reversed versions of each stream, which in parts present their own unique scenario. For example, reversing the “similarity” stream results in highly dissimilar concepts present within the first updates, with a convergence of concepts towards the end. As it turns out, this vastly impacts the trajectory, performing more consistently than even randomised streaming (https://bit.ly/3SUQnE1). Importantly, even with all the additionally included data streams, we find further corroborative evidence that if update cycles operate over the same data distribution, the trajectories are very likely to converge towards comparable performance end points.
>
> 4. ***New Methods (GaLore):***
> We have included GaLore (Zhao et al.) for gradient-based rank approximations to incorporate another conceptual approach to continual pretraining, and find an interesting accumulation-retention tradeoff behaviour, providing a simple middle-ground between finetuning and parameter-efficient finetuning. https://bit.ly/3WN6a93.
>
> > No new dataset introduced
>
> Beyond the construction, testing and aggregation of all components going into FoMo-in-Flux, we wish to highlight two aspects:
> - We note that a dataset is a combination of *data points*, e.g., images/videos, and *annotations*, e.g. class labels, bounding boxes, segmentation masks, or captions (we refer to sec 2.1 of [A survey on dataset quality in machine learning
> ](https://www.sciencedirect.com/science/article/pii/S0950584923001222)). For example, we point the reviewer to the SugarCREPE dataset (which introduces new captions over COCO images, it was also published at NeurIPS D&B in 2023) and COCO-ReM (which introduces new segmentation masks over COCO images), both of which reuse images from the COCO dataset, while adding a significant layer of new annotations and metadata. While FoMo-in-Flux reuses images  from existing datasets, we enhance them with a completely new layer of annotation. Each of the over 2M images in our benchmark is augmented with a high-quality caption generated through the two-stage pipeline described in Section 2. Without this step, most  datasets included in our benchmark **would not be suitable** for contrastive vision-language (pre-)training, since no meaningful captions are available and class labels are not sufficient. This is a novel contribution on the dataset side, and we believe it will be beneficial for practitioners even beyond continual pretraining studies. For the  first time, it allows to investigate highly controlled continual pretraining of vision-language encoder models, which also aligns with the NeurIPS call for ‘Data-centric AI methods to measure and improve data quality or utility that bring important new insight.’
> - In addition, we also introduce two synthetic datasets created entirely from scratch, ObscureAnimals and ObscureThings. These were generated using GPT-4 and state-of-the-art text-to-image models, followed by careful manual curation to ensure quality and accuracy, with the goal of incorporating long-tail semantic concepts as well as generative data into the benchmark as well. Download links for both datasets are also provided in the paper.

---

> > ### Author Response · Authors · 2024-08-17
> > **Rebuttal contd.**
> >
> > > Dataset combination limit the potential multimodal application.
> >
> > We respectfully disagree with the reviewer’s assertion: dataset mixtures are a crucial element in the development of strong multimodal training pipelines. This is evident from several prominent works, including DataComp, PaLI-Gemma, and LocCA, which emphasise the importance of using diverse and comprehensive dataset combinations to enhance model performance/generalization. Specifically, the inclusion of varied datasets allows models to better capture complex relationships between modalities, such as image and text, leading to robust learning outcomes.
> > Concretely, our improved dataset combination directly enables researchers to incorporate them into their training mixtures for tasks like the BYOD track in DataComp. Cleaner, large-scale, fine-grained captions and richer dataset combinations have been central to building stronger, more generalizable models. We believe this contribution is essential for advancing the field of multimodal learning and addressing the challenges associated with continual pretraining.
> >
> > > The insights provided with pretraining seems to be limited and may lack a broader audience due to the choice of the model and diversity of the benchmark.
> >
> > We hope that the addition of compute and model scaling insights, alongside extended experiments have helped address this concern.
> >
> > ***Choice of models.*** We highlight that our backbone(s) are widely used and follow standard literature (e.g. TiC-CLIP, OpenCLIP): see also L176-182 in the supplementary. The supplementary also shows consistent results across different patch sizes.
> >
> > ***Diversity of benchmark.*** While our benchmark is highly diverse (see Supp. Tab. 2,Tab. 3 showcasing our full set of datasets) and covers multiple, often disjoint domains, the data stream ordering allows for controlling the nature and magnitude of distribution shifts. We believe this makes our benchmark flexible and widely usable, unlocking new directions in continual pretraining.
> >
> > > The authors didn't mention any limitations and potential negative societal impact of their work.
> >
> > Thank you for this feedback. We kindly refer to our supplementary, which provides a detailed discussion on limitations, societal impact, and future outlook of our work. We will ensure the final draft refers this more clearly.
> >
> > >  the fonts of the figures can be improved.
> >
> > We thank the reviewer for pointing this out, and will increase font sizes for figures in the final draft.
> >
> > > it lacks significant difference if compared to some other existing work
> >
> > We respectfully disagree on the lack of significant differences with respect to existing work, as we make multiple novel contributions---both in the design and construction of the benchmark, our perspective on continual pretraining and the large array of application-relevant studies. More precisely:
> > - We introduce a new large-scale, multimodal benchmark over multiple visual/semantic domains, enabling new avenues into continual multimodal pre-training. We provide high-quality captions for the ~2M images in our benchmark, and release two image-text datasets created entirely from scratch (Obscure Things,Obscure Animals). We compare our contribution to existing work in Tab 1, showing that it offers a unique, configurable resource to study continual pretraining. We added two more axes of comparison to Tab 1 (see link in shared response) for even better separation: "Data-Mixtures"- the ability to easily mix and merge different sources of pretraining data. "Real World Stream Variations"- the ability to flexibly design data stream orders mimicking potential deployment scenarios.
> > - In addition, we release training code that allows researchers to easily define and test new continual pre-training scenarios using flexible and easy to extend configuration files (also provided in our submission).
> > - Thanks to the concept-first design, we are able to go beyond traditional dataset-centric approaches and study data streams that closely resemble real-world scenarios, offering an entirely new perspective on continual pre-training.
> > - We introduce a novel conceptual framework of continual learning in foundation models and highlight a previously underexplored setting that lies between large-scale pre-training from scratch and local, isolated knowledge edits.
> > - We study this setting through various lenses: we test different continual learning methods, stream orderings, learning rate schedules, and data mixtures. Based on feedback from the reviews, we also perform a number of additional experiments, detailed in the shared response, including an investigation of model and compute scaling, additional data streams, and new learning rate schedules. We hope that our findings can serve as a detailed and practical reference for practitioners and the benchmark and training codebase is a useful academic resource, unlocking new research into continual pre-training of foundation models.

---

> > > ### Comment · Reviewer_4Xf5 · 2024-08-19
> > >
> > > Thanks for the response and additional work. Based on the additional work, this submission is more solid and provides a greater contributions. However, based on the additional experiments listed in the response, the effects of the model size etc. can have a very large impact. Which let people wondering what happens when more backbones are tested. I have upgrade my score to reflect the new experiments.

---

### Author Rebuttal · Authors · 2024-08-17

We would like to thank all the reviewers for their feedback, and in particular their appreciation of the **overall high quality (4Xf5, jRa8)**, **originality and novelty (4Xf5, jRa8)**, and **good organisation and clarity (4Xf5, jRa8, Fxhw)** of our submission, as well as our **extensive and comprehensive benchmark and experiments (jRa8, Fxhw)**, and the **valuable, highly applicable insights and references for the research community (4Xf5, jRa8, Fxhw)**.

For each reviewer, we have addressed individual questions and comments under their respective review. In this shared response, we provide a summary of changes, and the additional experiments and insights that we have included:

**Textual changes:**
- Better clarified the use of continual pretraining to mean the general task of continuously updating large-scale pretrained foundation models, and fine-tuning to mean the specific method of naive continuous training using the CLIP pretraining objective (i.e., fine-tuning with the image-text contrastive loss).
- Provided pointers to detailed comparison with existing works in the supplementary and more clearly characterised and differentiated Fomo-in-Flux in Tab.1
- Discussed the limitations and societal impact of our work in more detail in the main paper.

**Experiments:**
- ***Model and Compute Scaling Experiments, and extension of our base experiments:*** Following the feedback from Reviewers 4Xf5 and jRa8, we have provided additional experimental axes to further extend the generality and relevance of our insights---on model scaling, compute scaling and extension of most of our base experiments.
  - *Model Scaling:* We show that scaling up the model size (from OpenCLIP ViT-S/16 with 62.3M parameters to OpenCLIP ViT-g/14 with 1.37B parameters) actually facilitates knowledge accumulation at improved zero-shot retention rates! This means that from a purely long horizon deployment scenario, larger models are more amenable to continuous model updates. [Link to result plots](https://bit.ly/4fRf7H6)
  - *Compute Scaling:* On the other hand, we find that simply increasing the associated compute budget does not immediately benefit the accumulation-retention tradeoff for naive continuous fine-tuning. However, in combination with model merging, larger compute budgets can improve the accumulation-retention tradeoff to a certain extent, exhibiting a log-linear compute-tradeoff scaling. [Link to result plots](https://bit.ly/4ctYSgg)
  - *Consistency over Patch Resolution:* Moreover, we highlight that the ablation experiments in the supplementary (figure 2 center in the supplementary) show that our insights also hold for changes in patch size when moving from patch-size 16 to 32 (i.e. ViT-B/16 → ViT-B/32).
  - *Additional datastreams:*
    - We have included two additional dataset-level streams: dataset-incremental, and a NEVIS-inspired time-incremental dataset ordering. In both cases, we find comparable behaviour, namely high trajectory variance attributable to the expected high visual biases within each dataset (c.f. Torralba et al. 2011 “Unbiased look at dataset bias”, Liu et al. 2024 “A Decade’s Battle on Dataset Bias”). [Link to result plots](https://bit.ly/4cpvNmg)
    - Moreover, we incorporate reversed versions of each stream, which in parts present their own unique scenario. For example, reversing the “similarity” stream results in highly dissimilar concepts present within the first set of updates, with a convergence of concepts towards the end. As it turns out, this significantly impacts the trajectory, performing more consistently than even random streaming. [Link to result plots](https://bit.ly/3SUQnE1)
    - With all the additionally included data streams, we find further evidence that as long as update cycles operate over the same data distribution, the continual training trajectories are very likely to converge towards comparable performance end points.
  - *Additional Meta LR Schedule and continual pretraining method:*
    - For completeness, we have included a dynamics-recovering meta cosine lr-scheduler in our meta-scheduler study (see Cosine Continued Dynamic [here](https://bit.ly/4cs6jod), as well as GaLore in our method study to also incorporate recent approaches leveraging gradient-based rank approximations, see [here](https://bit.ly/3WN6a93).
  - Additional Discussion of the importance of low, aligned softmax temperature choices in continual pretraining based on [these experimental visualizations](https://bit.ly/3YOeojT).
- ***Expanded Related Works / Benchmark comparison in Tab. 1.***
  - We have moved section 2.1 into its own unique section, to more clearly highlight that it is both our discussion of related works, and a new categorization of continual pretraining within the existing literature. Moreover, we have provided additional pointers to our discussion and detailed benchmark comparison with TiC-CLIP and NEVIS in the supplementary section 3.
  - We have added two more axes to Table 1 (see [here](https://bit.ly/4dJGbX2)), both of which even more clearly separate FoMo-in-Flux from existing works. *"Data-Mixtures"* highlights the ability to easily mix and merge different pretraining data sources and *"Real World Stream Variations"* describes the ability to construct, design and test different controlled data stream settings mimicking potential deployment scenarios. The latter in particular is only possible due to our *concept-first benchmark design*, and the corresponding *generation of much higher quality, better aligned image captions through our two-stage system*.

---

### Decision · Program_Chairs · 2024-09-26

**Decision:**

Accept (Poster)

**Comment:**

[Paper Summary]
The paper introduces a benchmark for continual multimodal learning by combining 63 datasets. The process of combining these datasets includes enhancing captions and collecting new data. The paper also presents experimental results using various learning strategies applied to the benchmark.
[Assessment of Reviews]
There was some disagreements among the reviewers regarding the novelty of the paper. Reviewer jRa8 stated that the benchmark is a novel contribution, while Reviewer 4Xf5 argued that it lacks significant difference when compared to other existing work.
All reviewers agreed that the paper is easy to follow and includes diverse experimental results.
The authors responded to Reviewer 4Xf5's concerns by elaborating on the significance of their contributions. The authors also addressed other reviewers' comments regarding opportunities for improvement and limitations.
[Justification for Decision]
Based on recommendations from the reviewers (7 (good paper, accept), 8 (top 50% of accepted papers, clear accept), 7 (good paper, accept)), I recommend accepting this the paper.
[Comment]
I recommend including a more detailed description of the contribution regarding the dataset in the abstract for the camera-ready version. This will improve the reader's understanding the content and flow of the paper.